# Reading the Cell, Designing the Cure: Perturbation-Conditioned Molecular Diffusion for Function-Oriented Drug Design

**Ziyu Xu** [* 1 2]  **Zijian Zhang** [* 1 2]  **Liang Wang** [1 3]  **Zhiyuan Liu** [4]  **Qiang Liu** [† 1 3]  **Shu Wu** [1 3]  **Liang Wang** [1 3]

## Abstract

When reliable target structures are unavailable at scale or phenotypes arise from dysregulated pathways, transcriptomic perturbations provide a system-level functional readout for drug action. In this work, we formalize *Transcriptome-based Drug Design (TBDD)* as a generative inverse problem: designing drug molecules conditioned on desired transcriptomic state transitions. We analyze the inherently ill-posed nature of this task, which is further complicated by the profound domain gap between biology and chemistry and by the sparsity of transcriptomic signals. To address these challenges, we propose **CURE** (A **C**ell**U**lar **R**esponse **E**ngine), a multi-resolution transcriptome-guided diffusion framework. CURE features a specialized **Transcriptome Perturbation Functional Feature Extractor (TFE)** that (1) distills function-oriented perturbation embeddings from pre/post states, (2) aligns these signatures to dual chemical views to bridge the cross-modal gap, and (3) performs heterogeneity-aware aggregation to extract robust state-specific signals from noisy transcriptomic data. Extensive evaluations on both standard benchmarks and rigorous out-of-distribution protocols demonstrate that CURE consistently outperforms strong baselines in structural quality and functional consistency. Furthermore, we validate its practical utility via a zero-shot gene-inhibitor design task, highlighting the potential of phenotype-driven generative discovery.

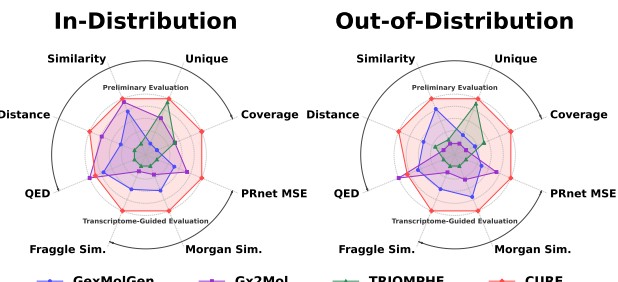

**In-Distribution**  **Out-of-Distribution**

GexMolGen — Gx2Mol — TRIOMPHE — CURE

*Figure 1.* **Performance overview across diverse evaluation metrics.** CURE achieves strong overall performance across structural metrics and function-consistency proxies in both in-distribution and out-of-distribution settings.

## 1. Introduction

Drug discovery remains a costly and failure-prone process (Sadybekov & Katritch, 2023). While computational pipelines have long aimed to accelerate this trajectory, the field has been predominantly governed by *Structure-Based Drug Design (SBDD)* (Bai et al., 2024; Saini et al., 2025). SBDD relies on the lock-and-key principle, utilizing three-dimensional (3D) protein target structures to design high-affinity ligands. However, this reductionist paradigm faces inherent bottlenecks: it falters when target structures are unknown (e.g., disordered proteins) or when disease phenotypes emerge from dysregulated multi-pathway networks rather than a single actionable target (Munson et al., 2024). Consequently, there is an urgent need for a complementary, *function-oriented* design paradigm that can bypass explicit target structural constraints and directly address cellular phenotypic shifts.

Transcriptomic perturbation signatures offer precisely this functional blueprint. Unlike static structural data, the transition from a pre-perturbation state to a post-perturbation state (i.e., $T_{pre} \rightarrow T_{post}$) captures the global functional impact of a molecule on a cellular system (Bunne et al., 2024; Ji et al., 2021). This differential profile integrates pathway-level interactions and network effects, effectively encoding the *molecule's mechanism of action (MoA)*. Despite the richness of this data, existing transcriptomics-driven machine learning methods predominantly address the *forward* problem: *predicting cellular responses to known compounds* (Hsieh et al., 2023; Wei et al., 2022). This asymmetry leaves the full potential of perturbation data untapped.

---

[*]Equal contribution  [1]NLPR, MAIS, Institute of Automation, Chinese Academy of Sciences, Beijing, China [2]School of Advanced Interdisciplinary Sciences, University of Chinese Academy of Sciences, Beijing, China [3]School of Artificial Intelligence, University of Chinese Academy of Sciences, Beijing, China [4]National University of Singapore, Singapore. Correspondence to: Qiang Liu <qiang.liu@nlpr.ia.ac.cn>.

*Proceedings of the 43rd International Conference on Machine Learning*, Seoul, South Korea. PMLR 306, 2026. Copyright 2026 by the author(s).

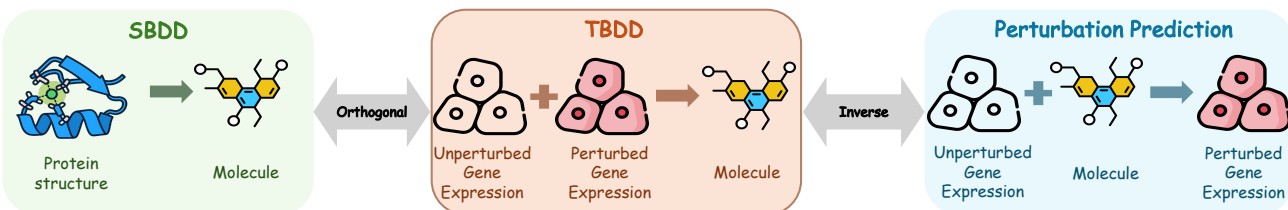

*Figure 2.* **Schematic illustration of TBDD and its relationships with existing paradigms.** TBDD is the *reverse* (design) direction complementary to *Perturbation Prediction*, and serves as a function-oriented complement to SBDD.

We argue that to truly complement SBDD, we must invert this workflow: leveraging phenotypic signatures not as prediction targets, but as *generative conditions* to guide the design of molecules that induce desired functional states (Figure 2).

To this end, we focus on *Transcriptome-based Drug Design (TBDD)*. Although preliminary explorations have touched upon this question, the field lacks a rigorous problem formulation and a systematic evaluation framework. We formalize TBDD as an inverse problem: given a target functional transition $(\mathbf{T}_{\text{pre}}, \mathbf{T}_{\text{post}})$ representing a therapeutic goal, the objective is to learn a conditional generator $p(\mathbf{G} \mid \mathbf{T}_{\text{pre}}, \mathbf{T}_{\text{post}})$ over drug molecules. This setting is (i) *orthogonal* to SBDD, conditioning on functional outcomes rather than physical constraints, and (ii) *inverse* to perturbation prediction. Crucially, TBDD is inherently ill-posed: transcriptomes encode functional effects rather than a unique atomic blueprint, and many distinct structures can yield similar signatures. We embrace this reality with a distributional view: instead of seeking a unique inverse, we aim to sample diverse, *functionally consistent candidates*.

Despite its promise, three challenges make TBDD difficult in practice. **(1) Cross-modality domain gap:** transcriptomic profiles and molecular graphs differ fundamentally in information density and inductive biases, making naïve direct conditioning unstable (Xiao et al., 2024; Zhou et al., 2025). **(2) Sparse, noisy single-cell signals:** single-cell RNA-seq offers access to heterogeneous drug responses, but dropout, batch effects, and high-dimensional noise make conditioning brittle. Meanwhile, compatibility with bulk transcriptomics is essential to exploit vast, high-value legacy datasets (Hafemeister & Halbritter, 2023; Van de Sande et al., 2023). **(3) Evaluation under limited ground truth:** large-scale wet-lab validation is expensive, requiring careful proxy evaluation, strong retrieval baselines, and audit-friendly split protocols to mitigate leakage and memorization concerns.

To address these challenges, we present **CURE** (A **C**ell**U**lar **R**esponse **E**ngine for Transcriptome-based Drug Design), a multi-resolution transcriptome-guided diffusion framework for *de novo* molecular generation. CURE introduces a Transcriptome Perturbation Functional Feature Extractor **(TFE)** that (i) distills a function-oriented perturbation

embedding via the Bidirectional Transcriptome Perturbation Signal Interaction module **(TFE-I)** and maps it into *dual-view aligned chemical domains* (graph-topology and fingerprint views) through the Dual-View Molecular Domain Alignment module **(TFE-A)**; and (ii) leverages sparse scRNA-seq with the Heterogeneity-Aware Transcriptome Aggregation module **(TFE-H)** to suppress technical noise while preserving subpopulation variation. Finally, CURE employs a Graph Diffusion Transformer as the generative backbone, which iteratively reconstructs molecular graphs by conditioning on the extracted perturbation representations via Adaptive Layer Normalization (AdaLN).

Across multiple datasets and evaluation axes (distributional quality, structural sanity/diversity, and function-consistency proxies assessed by independent perturbation estimators), CURE consistently outperforms strong baselines (Figure 1). We further showcase a zero-shot gene-inhibitor design scenario, illustrating the practical utility of transcriptome-guided generation. Our contributions are as follows:

- We **formalize the task** of TBDD and provide a **systematic analysis** of its unique challenges.

- We propose **CURE**, a multi-resolution diffusion framework that enables robust conditioning by aligning functional signals with chemical domains and suppressing noise in sparse transcriptomic data.

- We design a **comprehensive evaluation suite** incorporating rigorous out-of-distribution and zero-shot protocols, demonstrating CURE's consistent superiority over baselines in both structural and functional metrics.

**Conflict of Interest Disclosure.** The authors declare no financial conflicts of interest related to the work presented in this paper.

## 2. Related Work

**Machine-Learning–Based Molecular Design.** Deep molecular design has evolved from SMILES sequence models to graph-based approaches that preserve molecular topology (Wang et al., 2025; Gómez-Bombarelli et al., 2018; Hu et al., 2025). Hierarchical generators such as (Jin et al., 2020; You et al., 2024; Weller & Rohs, 2024) efficiently

construct large molecules in a coarse-to-fine manner. Yet unconditional generation is unfocused for drug-design goals. Transformer-based graph diffusion models (Liu et al., 2024; Peng et al., 2023; Hoogeboom et al., 2022; Schneuing et al., 2024) enable multi-conditional generation via mechanisms like AdaLN to inject external signals. *Structure-based drug design* (SBDD) remains a classical conditional paradigm that uses 3D pocket structures to guide ligand generation (Alakhdar et al., 2024; Guan et al., 2024), but its single-target perspective limits performance on multi-pathway diseases and relies on high-quality protein structures (Isert et al., 2023; Wang et al., 2018; Fahim, 2025).

**Cellular-Perturbation Transcriptomics.** Transcriptomics offers a comprehensive snapshot of cellular function. Large perturbational resources, such as (Subramanian et al., 2017; Gao et al., 2019; Zhang et al., 2025), provide massive gene-expression profiles under chemical or genetic perturbations. Building upon them, predictive models (Qi et al., 2024; Hetzel et al., 2022; Lotfollahi et al., 2019; Roohani et al., 2024) integrate chemistry and baseline state to forecast single-cell or bulk responses, while frameworks like (Adduri et al., 2025) target heterogeneity and batch effects. Although useful for simulating responses, such models are predictive rather than generative. Emerging *transcriptome-guided generation* methods (Li & Yamanishi, 2025; Kaitoh & Yamanishi, 2021; Cheng et al., 2024) depend on explicit statistics that risk losing information, and they still face the ill-posedness of mapping macroscopic signals to complete structures. These issues underline the need for function-centric conditioning and architectural decomposition, which we pursue in CURE.

## 3. Setting and Problem Formulation

We consider three spaces. The **chemical space** $\mathcal{G}$ contains molecules represented as attributed graphs $\mathbf{G} = (\mathcal{V}, \mathcal{E})$. The **transcriptome space** $\mathcal{T} \subset \mathbb{R}^d$ contains gene-expression states $\mathbf{T} \in \mathbb{R}^d$, where $d$ is the number of measured genes (bulk) or a harmonized feature dimension (single-cell).

A **perturbation signature** can be specified as $(\mathbf{T}_{\mathrm{pre}}, \mathbf{T}_{\mathrm{post}}) \in \mathcal{T} \times \mathcal{T}$ or a derived representation $\mathbf{z} = g(\mathbf{T}_{\mathrm{pre}}, \mathbf{T}_{\mathrm{post}})$ (e.g., log-fold change or learned embeddings). Given optional cellular context $c$ (cell type, state, batch, etc.), the goal of TBDD is to learn a conditional distribution over molecules

$$p(\mathbf{G} \mid \mathbf{z}, c) = p(\mathbf{G} \mid \mathbf{T}_{\mathrm{pre}}, \mathbf{T}_{\mathrm{post}}, c), \quad (1)$$

from which we can sample candidate molecules whose induced cellular responses are functionally consistent with the target signature.

## 4. Multi-Resolution Transcriptome-Guided Diffusion Model

### 4.1. Model Architecture

Our proposed CURE method constructs a graph diffusion model based on transcriptome perturbation signals in gene expression profiles for controlled molecule generation. This model consists of two main parts: a **Transcriptome perturbation Functional feature Extractor (TFE)** and a **Perturbation feature-guided Molecular graph Diffusion model (PMD)**. The TFE fuses transcriptome information before and after perturbation and aligns it with the drug molecule feature space. PMD guides drug molecule generation by injecting perturbation signals into the conditional diffusion process. CURE is the first drug molecule generation method to integrate multi-resolution cellular perturbation data while preserving heterogeneity information. Furthermore, the generated molecules can be directly used for various downstream tasks, such as gene inhibitor discovery (Figure 3).

### 4.2. Perturbation Feature-Guided Molecular Graph Diffusion Model

We used a conditional molecular generation diffusion model guided by the perturbation representations from the TFE. The core architecture is based on the Diffusion Transformer (Peebles & Xie, 2023), where the conditional representations are injected to guide the denoising process.

**Molecular Graph Diffusion Model.** The graph diffusion model uses a Markov chain-driven forward process to progressively add noise to the molecular graph's discrete features (atom and bond types):

$$q\left(X_G^t \mid X_G^{t-1}\right) = \mathrm{Cat}\left(X_G^t; \tilde{p} = X_G^{t-1}\mathbf{Q}_G^t\right), \quad (2)$$

where $X$ is the matrix representing the graph $G$ and $\mathbf{Q}$ is the graph transition matrix. A neural network-parameterized reverse process can reconstruct the graph from noise by iteratively removing it. The reverse process learns to predict the original graph:

$$p_\theta\left(\tilde{G}^0 \mid G^t\right) = \prod_{t \in T} p_\theta\left(G^{t-1} \mid G^t\right). \quad (3)$$

$p_\theta\left(\tilde{G}^0 \mid G^t\right)$ is combined with $q\left(G^{t-1} \mid G^t, G^0\right)$ to predict the graph reverse distribution:

$$p_\theta\left(G^{t-1} \mid G^t\right) = q\left(G^{t-1} \mid \tilde{G}, G^t\right) p_\theta\left(\tilde{G} \mid G^t\right). \quad (4)$$

The training objective is to minimize the negative log-likelihood:

$$\mathcal{L} = \mathbb{E}_{q(G^0)}\mathbb{E}_{q(G^t|G^0)}\left[-\mathbb{E}_{\mathbf{x} \in G^0} \log p_\theta\left(\mathbf{x} \mid G^t\right)\right]. \quad (5)$$

**Transcriptome Perturbation Conditioned Molecular Generation.** The biodomains transcriptome perturbation

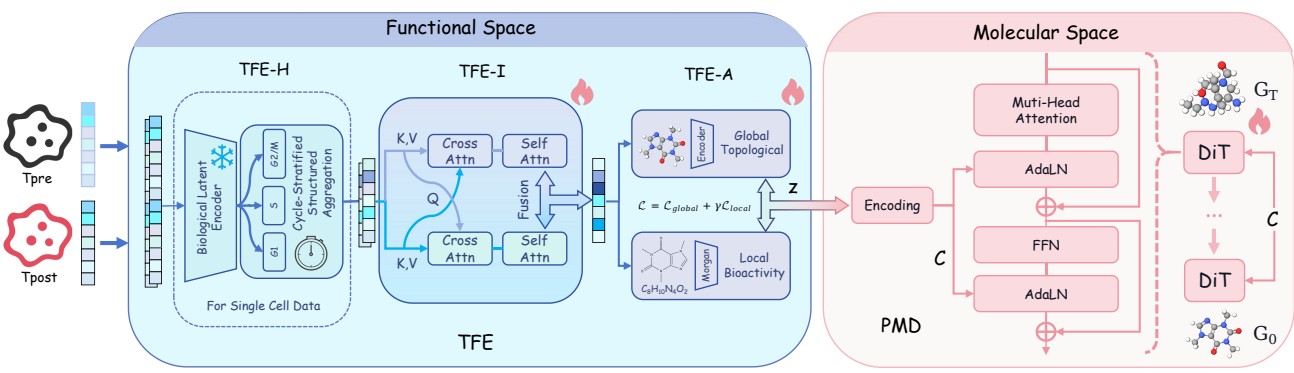

*Figure 3.* Overall architecture of CURE. The model consists of a Transcriptome Perturbation Functional Feature Extractor (TFE) that processes transcriptome expression data ($\mathbf{T}_{\text{pre}}$, $\mathbf{T}_{\text{post}}$) to produce a conditional embedding ($\mathbf{C}$), and Perturbation feature-guided Molecular graph Diffusion model (PMD) uses the condition to generate a target molecule.

representation from the TFE is injected into the Molecular with AdaLN method, guided by a multidimensional cluster embedder. We use Classifier-Free Guidance (CFG) (Ho & Salimans, 2022) to implement conditional generation:

$$\hat{p}_\theta \left( G^{t-1} \mid G^t, \mathbf{C} \right) = \log p_\theta \left( G^{t-1} \mid G^t \right)$$
$$+ \mathbf{s} \left( \log p_\theta \left( G^{t-1} \mid G^t, \mathbf{C} \right) - \log p_\theta \left( G^{t-1} \mid G^t \right) \right), \quad (6)$$

where $\mathbf{s}$ represents the scale of guidance and $\mathbf{C}$ represents the condition. During training, we use dynamic feature dropping and noise injection:

$$\mathbf{C} = \begin{cases} E_\theta(\mathbf{z}^t) + \boldsymbol{\epsilon} & \text{with } 1 - p \\ \mathbf{e}_{\text{drop}} + \boldsymbol{\epsilon} & \text{with } p \end{cases}, \quad \boldsymbol{\epsilon} \sim \mathcal{N}(0, \mathbf{I}). \quad (7)$$

With probability $p$, the embedding $E_\theta$ is replaced by a learnable dropout vector $\mathbf{e}_{drop}$; otherwise, it is processed by embedder $E_\theta$. Isotropic noise $\boldsymbol{\epsilon}$ is then added.

### 4.3. Transcriptome Perturbation Functional Feature Extractor

To efficiently extract perturbation functional signals in the biological transcriptome space for molecular condition generation, we designed a **Transcriptome Perturbation Functional Feature Extractor (TFE)** using methods of heterogeneity information preservation, perturbation signal interaction, and molecular domain alignment. As illustrated in Figure 3, the TFE comprises three parts: a **Heterogeneity-Aware Transcriptome Aggregation module (TFE-H)**, a **Bidirectional Transcriptome Perturbation Signal Interaction Module (TFE-I)**, and a **Dual-View Molecular Domain Alignment Module (TFE-A)**. All modules are progressively integrated to extract transcriptome perturbation functional signals and align them to drug molecule domains.

Compared to existing methods, CURE can perform feature interaction and extraction on both bulk and single-cell data while preserving heterogeneous information. To achieve

heterogeneous information preservation in single-cell transcriptome data, we specifically designed the TFE-H, which efficiently encodes single-cell data, thereby improving the information utilization rate of single-cell data. In the TFE-I, paired transcriptome perturbation data are interacted to extract the perturbation's biological functional signals. Finally, the functional perturbation signals are aligned with features of multiple molecular domains for subsequent conditionally controlled molecular generation tasks.

**Heterogeneity-Aware Transcriptome Aggregation.** To address the inherent trade-off between preserving population heterogeneity and mitigating sequencing noise when *conditioning* molecular generation on sparse single-cell data, we propose the TFE-H. Conventional approaches typically collapse complex cellular populations into a single mean vector, inevitably obscuring sub-population specific drug responses. In contrast, our method is designed to construct a robust, fine-grained representation of the cellular state distribution, ensuring that the generative process is conditioned on subtle, phenotype-driven perturbation signatures rather than homogenized signals. Specifically, we first leverage the SCimilarity (Heimberg et al., 2025) to project high-dimensional, sparse raw expression profiles into a dense, biologically rich latent manifold. Crucially, we implement a Cycle-Stratified Structured Aggregation strategy: we partition the population based on cell cycle phases (G1, S, G2/M) and transcriptional clusters, performing hierarchical sampling and local aggregation within these biologically coherent groups. This mechanism functions as a structured denoiser, effectively smoothing out random technical noise while rigorously preserving the distributional variance and sub-population heterogeneity essential for precise transcriptome-based drug design.

**Bidirectional Transcriptome Perturbation Signal Interaction** To distill the precise causal perturbation signature from the complex cellular background, we design the TFE-I, which employs a dual-stream architecture to explicitly

model functional shift between the pre- ($T_{pre}$) and post-perturbation ($T_{post}$) states. The TFE-I consists of three stacked interaction blocks, each maintaining separate processing streams for the unperturbed and perturbed representations derived from the TFE-H step. Within each block, we utilize a symmetrical Cross-Attention mechanism where $T_{pre}$ and $T_{post}$ reciprocally serve as queries to attend to each other's features. This design forces the model to dynamically align the two states and highlight the differential gene expression patterns driven by the drug. Following the cross-stream interaction, a Self-Attention layer within each stream refines the intra-state feature dependencies. Finally, the processed streams are integrated via an adaptive fusion unit, yielding a compact, function-oriented perturbation embedding that encodes the net therapeutic effect independent of basal cellular variations.

**Dual-View Molecular Domain Alignment.** The perturbation feature $z$, extracted by the TFE-I, resides in a continuous biological manifold. To effectively guide molecular generation, this feature should be mapped to a chemical space. However, a single chemical representation is often insufficient to capture the full complexity of drug-like molecules: graph embeddings excel at encoding global topology and validity, while molecular fingerprints are explicitly designed to capture local pharmacophores and bioactivity . To bridge this semantic gap and ensure the generated molecules are both structurally valid and functionally specific, we propose a Dual-View Molecular Domain Alignment module that projects the transcriptomic features into two complementary chemical domains.

**View 1: Global Topological Alignment.** To guarantee the structural validity of the generated molecules, we align the perturbation feature with the latent space of a pretrained hierarchical graph autoencoder (Jin et al., 2020). This latent manifold encapsulates essential chemical rules and topological constraints (e.g., valency and ring structures). By constraining the transcriptomic feature $z$ to map into this valid chemical space, we enforce the generative process to respect fundamental molecular topology. The global topological alignment objective is defined as:

$$\mathcal{L}_{\text{global}} = \mathcal{L}_{\text{ELBO}} + \mathcal{L}_{\text{align}}, \tag{8}$$

where $\mathcal{L}_{\text{ELBO}}$ is the standard VAE evidence lower bound loss, and $\mathcal{L}_{\text{align}}$ aligns the mean and variance of the transcriptome-derived features with the latent space.

**View 2: Local Bioactivity Alignment.** Global topology alone does not guarantee specific biological interactions. To explicitly encode the functional groups and pharmacophores, we introduce an alignment view using Morgan Fingerprints. Unlike graph embeddings, fingerprints provide a fixed-dimensional, sparse vectorization of local chemical environments, which are highly correlated with bioactiv-

ity targets. To map the continuous feature $z$ to this high-dimensional, sparse discrete space, we employ a Sparse-Aware Bioactivity Constraint. This objective fuses sparse regression with label-guided contrastive learning to handle the sparsity of the fingerprint space:

$$\mathcal{L}_{\text{local}} = \mathcal{L}_{\text{InfoNCE}} + \mathcal{L}_{\text{sparse}}. \tag{9}$$

By simultaneously minimizing $\mathcal{L} = \mathcal{L}_{\text{global}} + \gamma \mathcal{L}_{\text{local}}$, our model learns a unified latent representation that satisfies both the synthesizability constraints imposed by the graph domain and the functional specificity required by the fingerprint domain. This dual-view strategy effectively resolves the cross-modality domain gap by grounding the generation in chemically robust priors. Details are in Section A.5.

# 5. Experiments

In the experimental section, we follow the same perspective as our evaluation metrics, assessing the model's performance from three angles: macroscopic evaluation of the relationship between the generated and target molecular sets and the chemical and medicinal properties of the generated set itself, and microscopic evaluation of the effectiveness and accuracy of CURE conditional control generation. To demonstrate the model's generalization ability and the functional effects of the generated drugs, we designed the following three innovative evaluation experiments: zero-shot prediction of gene inhibitors, characterization of the functional effects of generated drugs, and accuracy assessment of the drug screener. To ensure reproducibility, we provide the necessary hyperparameter settings in Section D.1.

## 5.1. Experimental Setup

**Datasets.** *Bulk Cell Data*: We used the L1000 Level 3 dataset (Subramanian et al., 2017; Gao et al., 2019), which profiles the expression of 978 landmark genes across nearly 20,000 drugs and various cell lines. For training, we split the data 85:10:5 (train:test:val) using three strategies: random, mask drug, and mask cell. *Single-Cell Data*: We also utilized the Tahoe-100M dataset (Zhang et al., 2025), the largest single-cell perturbation dataset available. It contains results from over 300 drugs applied to 50 cancer cell lines, including their untreated states. *Gene Inhibitor Dataset*: For evaluation, we built a gene inhibitor dataset from the ExCape database (Sun et al., 2017). This set contains 1,200 to 23,000 known inhibitors for each of 10 selected human genes, enabling comparison with gene knockout expression profiles. To guarantee experimental fairness, we include a detailed description of the training data in Section D.3.

**Evaluation Metrics.** We used three types of metrics to assess the model's generative capabilities: *Macroscopic Metrics*: To reflect the properties of the entire generated set of drug molecules: (1) Heavy Atom Type Coverage (Cov-

*Table 1.* Comprehensive evaluation of CURE on both Bulk and Single-cell data. We report generalization performance and microscopic similarity across three generalization splits. To address the lack of dedicated single-cell baselines, we established a fair benchmark by adapting bulk models via pseudo-bulk profiling.

| Data Type | Split | Method | Preliminary Evaluation | | | | | Transcriptome-Guided Evaluation | | |
| --- | --- | --- | --- | --- | --- | --- | --- | --- | --- | --- |
| | | | Coverage ↑ | Unique ↑ | Similarity ↑ | Distance ↓ | QED ↑ | Fraggle Sim. ↑ | Morgan Sim. ↑ | PRnet MSE ↓ |
| Bulk | In-Distribution | GexMolGen | 54.55% | 0.7646 | 0.8919 | 35.4027 | 0.5127 | 0.7428 | 0.6939 | 4.6504 |
| | | Gx2Mol | 72.73% | 0.8360 | 0.9405 | 17.8963 | **0.6041** | 0.6203 | 0.5920 | 2.5987 |
| | | TRIOMPHE | 72.73% | 0.8809 | 0.7270 | 48.0169 | 0.3071 | 0.5842 | 0.5372 | 7.4599 |
| | | **CURE** | **100.00%** | **0.8906** | **0.9576** | **6.7856** | 0.5665 | **0.8892** | **0.8228** | **0.2328** |
| | Out-of-Distribution (Unseen Cells) | GexMolGen | 54.55% | 0.7622 | **0.8876** | 42.5445 | 0.5173 | 0.7285 | 0.6805 | 4.2724 |
| | | Gx2Mol | 45.45% | 0.7321 | 0.7123 | 65.9671 | **0.6203** | 0.6015 | 0.5849 | 3.7071 |
| | | TRIOMPHE | 63.64% | 0.8786 | 0.6637 | 56.0078 | 0.3209 | 0.5694 | 0.5138 | 8.6310 |
| | | **CURE** | **90.90%** | **0.8864** | 0.8238 | **13.6113** | 0.5736 | **0.9449** | **0.9125** | **0.2932** |
| | Out-of-Distribution (Unseen Drugs) | GexMolGen | 54.55% | 0.7609 | 0.9013 | 40.0122 | 0.5098 | 0.7194 | 0.6903 | 5.0482 |
| | | Gx2Mol | 45.45% | 0.7280 | 0.7106 | 64.6032 | **0.6232** | 0.6113 | 0.5882 | 2.7208 |
| | | TRIOMPHE | 63.64% | 0.8829 | 0.7324 | 54.3125 | 0.3589 | 0.5704 | 0.5081 | 7.4666 |
| | | **CURE** | **90.90%** | **0.9018** | **0.9576** | **9.5265** | 0.5725 | **0.8592** | **0.7722** | 0.4866 |
| Single-cell | In-Distribution | GexMolGen | 54.55% | 0.6521 | 0.5342 | 41.2201 | 0.3520 | 0.1988 | 0.2245 | 4.8549 |
| | | Gx2Mol | 45.45% | 0.7012 | 0.6102 | 61.4421 | 0.4541 | 0.2105 | 0.2884 | 4.1419 |
| | | TRIOMPHE | 63.64% | 0.7521 | 0.5002 | 55.2215 | 0.3984 | 0.1541 | 0.2011 | 7.7024 |
| | | **CURE** | **90.90%** | **0.8771** | **0.8137** | **27.6223** | **0.4946** | **0.7310** | **0.6114** | **0.4829** |
| | Out-of-Distribution (Unseen Cells) | GexMolGen | 54.55% | 0.5954 | 0.5769 | 47.6539 | 0.3362 | 0.2363 | 0.2621 | 5.1874 |
| | | Gx2Mol | 45.45% | 0.6758 | 0.4629 | 63.8786 | 0.4031 | 0.2130 | 0.2479 | 4.4892 |
| | | TRIOMPHE | 54.55% | 0.7310 | 0.4314 | 57.4050 | 0.3085 | 0.1430 | 0.1921 | 8.5492 |
| | | **CURE** | **90.90%** | **0.8474** | **0.7954** | **25.8473** | **0.4834** | **0.7023** | **0.6091** | **0.5392** |
| | Out-of-Distribution (Unseen Drugs) | GexMolGen | 54.55% | 0.5945 | 0.5858 | 46.0079 | 0.3313 | 0.1898 | 0.2280 | 5.6923 |
| | | Gx2Mol | 45.45% | 0.6732 | 0.4618 | 62.9920 | 0.4050 | 0.2429 | 0.2976 | 4.9385 |
| | | TRIOMPHE | 54.55% | 0.6738 | 0.4760 | 58.3031 | 0.3332 | 0.1535 | 0.2198 | 9.9462 |
| | | **CURE** | **90.90%** | **0.8361** | **0.7824** | **26.1922** | **0.4983** | **0.7164** | **0.6038** | **0.6482** |

erage); (2) Uniqueness of structures in a single generated batch (Unique); (3) Fragment-based similarity to a reference set (Similarity); (4) Fréchet ChemNet Distance to a reference set (Distance); (5) Quantitative Estimate of Drug-likeness (QED); (6) Synthesizability of the target molecule (SA); (7) Validity of generated molecules (Validity). *Microscopic Metrics*: To assess the reliability of drug prediction based on gene perturbation: (1) Fraggle-based molecular scaffold similarity (Fraggle Sim.); (2) Morgan fingerprint-based atomic environment similarity (Morgan Sim.) (Grant & Sit, 2021; Wang et al., 2022). *Experimental Design Metrics*: Innovatively designed to reflect the functional effects of generated drugs: (1) A metric to evaluate the difference in cellular gene expression effects between the generated drug and the ground-truth drug (PRnet MSE). (2) On zero-shot data of gene inhibitor effects, a metric to evaluate the similarity between the generated molecules and known gene inhibitors (Gene Inhibitor Sim.) (Méndez-Lucio et al., 2020).

**Baselines.** For the bulk data experiments, we selected several baseline models widely recognized in the TBDD task for comparison, including GexMolGen (Cheng et al., 2024), TRIOMPHE (Kaitoh & Yamanishi, 2021), and Gx2Mol (Li & Yamanishi, 2025). As for single-cell domain, there are currently no existing generative methods specialized for handling single-cell resolution inputs. To bridge this gap and establish a rigorous comparative benchmark, we adapted these strong bulk baselines by employing high-quality pseudo-bulk profiling, a standard computational biology technique that aggregates heterogeneous single-cell populations into a unified representation via averaging. Since these base-line architectures are inherently designed to process macroscopic transcriptomic signals, applying them to pseudo-bulked single-cell data constitutes a methodologically valid and logical comparison. Crucially, to ensure strict experimental fairness, all methods were optimized using identical underlying data sources: for bulk benchmarks, all models shared the same training splits; for single-cell benchmarks, baselines were trained on the exact same single-cell dataset processed via pseudo-bulk aggregation, directly contrasting with our model's ability to utilize the granular data.

**Evaluation Protocols.** To rigorously assess OOD generalization and rule out data leakage from training interpolation, we established two strict zero-shot protocols: **1) Unseen Cell Lines (Hierarchical Biological Split):** We implement a strict hierarchical separation across tumor types, tissues, and cell lines. By ensuring disjoint biological contexts between training and testing, this protocol challenges the model to disentangle intrinsic drug mechanisms from cellular heterogeneity, demonstrating robustness against transcriptomic background shifts. **2) Unseen Drugs (Scaffold-level Split):** We partition the dataset based on *Bemis-Murcko scaffolds* rather than random splitting. This strategy segregates distinct molecular frameworks to strictly prevent information leakage from structural analogs. Consequently, it compels the model to learn transferable structure-function mappings and valid pharmacophores, avoiding the pitfall of memorizing specific molecular templates (Appendix B).

*Table 2.* Zero-shot Gene inhibitor similarity and affinity.

| Target Gene | Morgan ↑ | | | | Affinity ↓ (kcal/mol) | | | |
|---|---|---|---|---|---|---|---|---|
| | Gex. | TRIO. | Gx2. | CURE | Gex. | TRIO. | Gx2. | CURE |
| AKT1 | 0.728 | 0.540 | 0.743 | **0.804** | -7.45 | -5.67 | -7.00 | **-8.63** |
| AKT2 | 0.712 | 0.515 | 0.706 | **0.754** | -7.48 | -5.82 | -7.39 | **-8.59** |
| AURKB | 0.744 | 0.553 | 0.719 | **0.760** | -7.40 | -6.49 | -7.38 | **-8.79** |
| CTSK | 0.749 | 0.535 | 0.699 | **0.751** | -7.55 | -6.38 | -7.29 | **-8.69** |
| EGFR | 0.747 | 0.540 | 0.738 | **0.782** | -7.30 | -6.11 | -6.91 | **-9.11** |
| HDAC1 | 0.720 | 0.519 | 0.697 | **0.772** | -7.00 | -5.85 | -7.27 | **-8.68** |
| MTOR | 0.794 | 0.527 | 0.745 | **0.808** | -7.09 | -5.91 | -7.02 | **-8.74** |
| PIK3CA | 0.764 | 0.524 | 0.726 | **0.809** | -7.33 | -5.80 | -7.47 | **-9.15** |
| SMAD3 | 0.845 | 0.590 | 0.843 | **0.881** | -7.28 | -5.99 | -7.35 | **-9.07** |
| TP53 | 0.809 | 0.588 | 0.793 | **0.816** | -7.09 | -5.76 | -7.43 | **-8.28** |

*Table 3.* TFE Performance of the drug screener.

| Top-K | Morgan Sim. ↑ | Hit Rate ↑ | PRnet MSE ↓ |
|---|---|---|---|
| 5 | **0.9753** | 0.6766 | 0.1668 |
| 10 | 0.9653 | 0.9192 | 0.1353 |
| 15 | 0.9568 | 0.9694 | 0.1228 |
| 20 | 0.9195 | **0.9790** | **0.1093** |

## 5.2. Preliminary Evaluation of Drug Generation

This experiment evaluates CURE from a macroscopic distributional perspective, verifying the quality, diversity, and chemical validity of the generated molecular space. As detailed in Table 1, our method establishes a new state-of-the-art across both Bulk and Single-cell modalities. A critical finding is that CURE achieves a remarkably high Unique score while maintaining the lowest Fréchet ChemNet Distance. This specific combination provides strong empirical support that the model performs true generative exploration, synthesizing novel, chemically valid structures, rather than merely reconstructing or memorizing the training scaffold. The superiority of our framework is most pronounced in the Single-cell setting. Despite our rigorous adaptation of baseline models using high-quality pseudo-bulk profiles to ensure a fair comparison, they suffer severe performance degradation due to signal dilution from naive averaging. In sharp contrast, CURE maintains robust distributional alignment. This empirically confirms that our TFE-H successfully extracts heterogeneity pharmacological signals from noisy cellular environments where conventional aggregation strategies fail, demonstrating robust adaptability even in the challenging Unseen Drugs split.

## 5.3. Evaluation of Transcriptome-guided Drug Molecular Generation

To rigorously evaluate conditional control, we assess performance via Structural Accuracy and Functional Fidelity. In terms of structure, CURE achieves near perfect alignment, vastly outperforming baselines. Even in the rigorous Unseen Drugs split, CURE maintains high similarity, indicating it has learned to identify and generate the essential pharma-

cophores and functional groups required to induce specific transcriptomic changes. In terms of function, to rigorously quantify biological efficacy in this target-free context, we introduced PRnet as a functional proxy. PRnet is a predictive model that takes the raw basal transcriptome and a drug molecule as inputs to predict the resultant drug-induced perturbation profile. We evaluate performance by calculating the MSE between the phenotypic vector predicted for our generated molecule and that of the ground truth; a lower MSE signifies closer functional proximity to the desired therapeutic effect. Crucially, to ensure the integrity of this metric, PRnet was trained on a strictly independent partition of the dataset to prevent data leakage. The result (Table 1) confirms that CURE achieves high functional consistency, generating candidates that effectively induce the target transcriptomic state.

## 5.4. Gene Inhibitor Prediction

To assess CURE's utility in drug development and validate its capability to capture functional biological mechanisms, we established a rigorous zero-shot benchmark targeting 10 canonical genes backed by extensive inhibitor libraries (Sun et al., 2017). To ensure a fair comparison, we enforced a strict protocol where models trained exclusively on standard drug-perturbation transcriptomes were tasked with generating molecules conditioned on unseen gene knockout (KO) signatures. We leverage KO profiles as "phenotypic anchors" representing the desired loss-of-function state, positing that a functionally aware model should generate structures that not only resemble known inhibitors but also exhibit physical binding potential to the target proteins. Consequently, beyond calculating structural similarity (Morgan Fingerprint), we further performed molecular docking to assess the average Binding Affinity (kcal/mol) of the generated candidates against the target protein structures.

As detailed in Table 2, CURE significantly outperforms all baselines across both evaluation dimensions. In terms of chemical structure, our method consistently achieves the highest similarity scores across all targets, indicating that the generated molecules share critical pharmacophores with known inhibitors. More importantly, in terms of physical interaction, CURE achieves the strongest binding affinities (lowest energy scores) compared to baselines. This consistency between high structural similarity and strong binding potential validates that CURE does not merely memorize chemical patterns but effectively extracts mechanism-specific signals from transcriptomic perturbations. By successfully translating phenotypic knockout signals into high-affinity inhibitor structures without explicit protein structure training, CURE demonstrates robust potential for function-oriented de novo drug design. Details are in Section A.4.

*Table 4.* Ablation study of the TFE. *Note: w/o TFE is trained with L1000 level 5.*

| Dataset | Module | | | | Validity↑ | Coverage↑ | Diversity↑ | Distance↓ | SA↓ | QED↑ | Morgan Sim.↑ |
|---|---|---|---|---|---|---|---|---|---|---|---|
| | TFE-H | TFE-I | TFE-A | | | | | | | | |
| | | | Global | Local | | | | | | | |
| L1000 | × | × | × | × | 0.8775 | 90.91% | 0.7504 | 8.4982 | 0.8355 | 0.5426 | 0.1824 |
| | × | ✓ | × | × | 0.3000 | 63.64% | 0.7662 | 82.7183 | 0.7651 | 0.4556 | 0.0886 |
| | × | × | ✓ | ✓ | 0.2400 | 36.36% | 0.6982 | 51.0830 | 0.6933 | 0.4400 | 0.2527 |
| | × | ✓ | ✓ | ✓ | **0.9350** | **100.00%** | **0.8906** | **6.7856** | **0.6386** | **0.5665** | **0.8228** |
| Tahoe | ✓ | ✓ | × | ✓ | 0.9650 | 81.82% | 0.8693 | 43.0227 | 0.6043 | 0.4588 | 0.2674 |
| | ✓ | ✓ | ✓ | × | 0.9400 | 81.82% | 0.8600 | 29.6223 | 0.6357 | 0.3048 | 0.3219 |
| | × | ✓ | ✓ | ✓ | 0.9450 | 90.91% | 0.8671 | 30.6223 | 0.6193 | 0.4645 | 0.4924 |
| | ✓ | ✓ | ✓ | ✓ | **0.9800** | **90.91%** | **0.8771** | **27.6223** | **0.5994** | **0.4946** | **0.6114** |

## 5.5. Performance Evaluation of TFE

To evaluate the effectiveness of TFE and verify the model's translational application value, we employed a drug screening method, using molecular features $z$ extracted by TFE as structural probes to query molecular fingerprints in a large-scale drug database. We performed a Top-K nearest neighbor search. Table 3 details the screening performance based on three metrics at different search thresholds $k$: the average structural similarity (Morgan) between the feature $z$ and the top $k$ retrieved candidate molecules, the probability that the ground truth drug appears among the retrieved candidate molecules (Hit Rate), and the functional difference in predicted gene expression effects between the retrieved candidate molecules and the ground truth drug (PRnet MSE). Increasing the $k$ value reveals a typical trade-off: although structural similarity naturally decreases due to the inclusion of distant neighbor molecules, search efficiency is significantly improved, reflected in higher Hit Rates and stronger functional alignments (lower PRnet MSE). Notably, when the threshold $k = 10$, the model achieved a Hit Rate more than 90%, demonstrating a powerful ability to identify target drugs based on functional transcriptome input. This demonstrates that CURE can refine a large chemical library into a clinically manageable library of compounds (e.g., 10-20 compounds) with high structure and function fidelity, providing a pragmatic solution for time-sensitive therapeutic applications. Details are in Section A.3.

## 5.6. Biochemical Interpretability Analysis

To systematically evaluate the interpretability of CURE, we analyzed the model's learned representations from two complementary dimensions: the biological relevance of the functional latent space and the chemical structural fidelity of the generated molecules.

**Biological Interpretability Analysis.** Since CURE relies on phenotypic changes without explicit affinity metrics, we employed stratified UMAP to verify mechanistic principles. First, projecting distinct inhibitors within fixed cellular backgrounds (Figure 6, top) revealed discrete clustering by inhibitor type. This topological separation implies the model encodes mechanism-specific signatures, mapping perturbations to MoA rather than fitting noise. Second, visualizing identical inhibitors across diverse cell lines (Figure 6, bottom) exhibited stratification by cellular identity, confirming the model dynamically adapts functional representations to biological contexts rather than overfitting. Collectively, these results demonstrate that the latent space effectively disentangles functional drug impacts from cellular backgrounds, supporting function oriented drug discovery.

**Chemical Structural Interpretability Analysis.** For the chemical structural analysis, we examined the generative diversity and structural logic of the output molecules through stochastic multi-sampling. We visualized multiple molecules generated from the same transcriptomic condition (as shown in Figure 5 and Table 7). The results indicate that while the generated molecules maintain high similarity scores (Fraggle/Morgan) to the reference drugs, they exhibit significant diversity in their SMILES representations. Generated molecules are not identical to training targets but share critical functional groups (pharmacophores) and local chemical environments. This confirms the model has learned the underlying mechanism of how specific chemical substructures drive transcriptomic changes.

## 5.7. Ablation Studies

We validated the contribution of each module as detailed in Table 4. Crucially, in the single-cell Tahoe dataset, removing the Heterogeneity Aware Aggregator (TFE-H) caused a marked decline in both distributional alignment and structural similarity. This confirms that TFE-H is indispensable for distilling robust perturbation signals from noisy, heterogeneous cellular data. Furthermore, the Interaction (TFE-I) and Alignment (TFE-A) modules proved foundational. Their exclusion on the L1000 dataset led to functional model collapse, with generated molecules losing validity. Specifically, the dual-view alignment is critical: removing local

fingerprint constraints resulted in a noticeable loss of pharmacophore fidelity, validating the necessity of our multi-domain alignment strategy for preserving biological activity.

## 6. Conclusion

In this work, we present CURE, a TBDD framework that bridges cross-modal gaps and transcriptomic noise via heterogeneity-aware aggregation and dual-view alignment. Our method effectively addresses the cross-modal domain gap inherent in target-free generation. Extensive validation confirms that CURE achieves superior structural accuracy and high functional consistency.

## Acknowledgments

This work was supported by the Strategic Priority Research Program of Chinese Academy of Sciences (Grant No. XDA0480102) and the National Science and Technology Major Project (2023ZD0120901).

## Impact Statement

This paper presents work whose goal is to advance the field of machine learning for drug discovery. The proposed framework generates candidate drug molecules conditioned on transcriptomic perturbation data, which could accelerate early-stage therapeutic discovery. While the generated molecules require extensive experimental validation before any clinical consideration, we acknowledge the dual-use potential inherent in generative molecular design. There are many potential societal consequences of our work, none of which we feel must be specifically highlighted here.

## Reproducibility Statement

To ensure reproducibility of our work, we have made our source code available at https://github.com/EdwardCurry/CURE_TBDD. Our experiments utilized exclusively open-access data, including the L1000 dataset (bulk RNA-seq) (Subramanian et al., 2017; Gao et al., 2019), Tahoe-100M (single-cell data) (Zhang et al., 2025), and ExCape (gene inhibitor information) (Sun et al., 2017). All hyperparameters used for training are explicitly documented in the configuration files within the code repository and in Section D.1. For detailed implementation and reproduction steps, please refer to the provided code and README documentation.

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

# A. Methodology Details

## A.1. Heterogeneity-Aware Transcriptome Aggregation Details

Current molecular generation tasks struggle to effectively utilize single-cell transcriptome data due to its high sparsity and significant technical noise. Naive aggregation methods (e.g., global averaging) tend to collapse cellular heterogeneity, causing the model to learn homogenized signals rather than robust biological responses. To address this, we designed the **Heterogeneity-Aware Transcriptome Aggregation (TFE-H)** module. This module functions as a *structured denoiser*, leveraging biological priors to smooth out technical noise while preserving fine-grained sub-population distributions. The process consists of two key stages:

1. **Manifold Projection via SCimilarity:** First, to mitigate the curse of dimensionality and sparsity dropout events ($D > 60,000$), we employ the SCimilarity framework to project raw expression profiles onto a biologically rich, dense latent manifold. SCimilarity is a deep metric-learning foundation model trained on approximately 7.9 million single-cell profiles across 56 studies (spanning multiple tissues and diseases). It maps transcriptionally similar cells to proximal points in a low-dimensional embedding space ($d = 128$). By using this pre-trained encoder, we perform *semantic compression*, extracting robust cell-state features that are resilient to technical noise before any aggregation takes place.

2. **Cycle-Stratified Structured Aggregation:** Instead of performing indiscriminate global pooling, we construct a *Structured Feature Matrix* to represent the cell population. For a given perturbation condition, we partition the $d$-dimensional cell embeddings into subsets based on their cell cycle phases (G1, S, G2/M). The cell cycle is compartmentalized into the G1 (Gap 1), S (Synthesis), and G2/M (Gap 2/Mitosis) phases, inherently encapsulates heterogeneity information derived from the specific sub-population variations and unique phenotype-driven drug responses distinct to each stage. To balance the signal-to-noise ratio, we perform hierarchical sampling: we first aggregate cells locally with pooling in these biologically coherent clusters to reduce variance, and then sample a fixed total of $N = 128$ representative feature vectors according to the population's cycle proportions with merging them to a distribution-level condition embedding. This approach ensures that the input to the diffusion model captures the full distributional shape of the cellular response, rather than a single mean vector.

**TFE-H Ablation Study.** To validate the necessity of our architectural design, we conducted comprehensive ablation studies on the TFE-H. To validate the theoretical premise of our heterogeneity-aware design, we conducted ablation studies comparing our Cycle-Stratified Structured Aggregation against a Naive Bulk Averaging baseline (where single-cell data is simply averaged into a pseudo-bulk vector). As shown in the ablation results (Table 5), the Naive Averaging strategy resulted in a significant drop in performance (e.g., increased Fréchet Distance), indicating that collapsing the population distribution leads to a loss of critical pharmacological signals. In contrast, our structured approach, which preserves cycle-specific variance and sub-population structures, yielded superior structural fidelity and functional alignment. This confirms that the performance gains of CURE stem from the explicit modeling of cellular heterogeneity, effectively bridging the resolution gap between noisy single-cell profiles and precise molecular generation. Furthermore, comparative analysis against MLP and scratch-trained CNN baselines (Table 5) confirms that TFE-H provides the essential inductive bias to extract information-dense features from these structured inputs, effectively bridging the modality gap between transcriptomic profiles and molecular structures.

## A.2. Formal Specification of TFE Modules

We provide the complete mathematical specification with tensor dimensions for each TFE sub-module.

**TFE-H (Heterogeneity-Aware Aggregation).** For single-cell input $X_{sc} \in \mathbb{R}^{n \times D}$ ($n$ cells, $D > 60,000$ genes):

1. *Manifold Projection:* $H = f_{\text{SCimilarity}}(X_{sc}) \in \mathbb{R}^{n \times 128}$, using a frozen pre-trained encoder.

2. *Cycle-Stratified Aggregation:* Partition $H$ into $\{H_{G1}, H_S, H_{G2/M}\}$ by cell cycle phase. Perform local pooling within each phase, then proportionally sample $N = 128$ representative vectors to form $T_{pre}, T_{post} \in \mathbb{R}^{128 \times d}$.

For bulk input, TFE-H is bypassed and $T_{pre}, T_{post} \in \mathbb{R}^{1 \times d}$ are used directly.

*Table 5.* Ablation study of TFE-H. We compared different architectures and strategies.

| Metric | w/o TFE-H | MLP | Conv | Naive Bulk Averaging | Ours |
|---|---|---|---|---|---|
| Coverage ↑ | 63.6% | 63.6% | 81.8% | 90.9% | **90.9%** |
| Unique ↑ | 0.45 | 0.79 | 0.84 | 0.86 | **0.88** |
| Similarity ↑ | 0.58 | 0.78 | 0.70 | 0.74 | **0.81** |
| Distance ↓ | 73.45 | 38.26 | 33.15 | 30.62 | **27.62** |
| QED ↑ | 0.39 | 0.44 | 0.42 | 0.46 | **0.49** |
| Fraggle Sim. ↑ | 0.53 | 0.58 | 0.63 | 0.64 | **0.73** |
| Morgan Sim. ↑ | 0.49 | 0.47 | 0.57 | 0.49 | **0.61** |

**TFE-I (Bidirectional Perturbation Signal Interaction).** Input: $T_{pre}, T_{post} \in \mathbb{R}^{N \times d}$. The module consists of 3 stacked interaction blocks, each applying symmetrical cross-attention followed by self-attention:

$$T'_{pre} = \text{SelfAttn}(\text{CrossAttn}(Q{=}T_{pre}, \ K{=}T_{post}, \ V{=}T_{post})),$$
$$T'_{post} = \text{SelfAttn}(\text{CrossAttn}(Q{=}T_{post}, \ K{=}T_{pre}, \ V{=}T_{pre})). \tag{10}$$

After 3 blocks, the two streams are concatenated and fused via self-attention followed by mean pooling to produce the perturbation representation $z \in \mathbb{R}^{d_z}$.

**TFE-A (Dual-View Molecular Domain Alignment).** *View 1 (Global):* A pre-trained HierVAE encoder $Q$ provides targets $(\mu_{enc}, \sigma_{enc}) = Q(X_G)$. TFE projects $z$ to $(\mu_f, \sigma_f) = g_{\text{proj}}(z)$, aligned via $\mathcal{L}_{\text{align}} = \|\mu_{enc} - \mu_f\|^2 + \|\sigma_{enc}^2 - \sigma_f^2\|^2$. *View 2 (Local):* A projection head produces $A = h_{\text{proj}}(z) \in \mathbb{R}^{2048}$ against target Morgan fingerprint $B = \text{MorganFP}(G) \in \mathbb{R}^{2048}$. The loss combines masked InfoNCE (positive pairs share the same SMILES label, off-diagonal same-label entries masked to $-\infty$, $\tau{=}0.1$) with sparse regression (non-zero positions weighted by $w = \log(1{+}B_{pos})$; zero positions penalized with $\alpha{=}0.4$, $\lambda{=}0.15$). Total: $\mathcal{L} = \mathcal{L}_{\text{global}} + \gamma \mathcal{L}_{\text{local}}$. Training protocol: TFE trained $\sim$30k steps, then frozen; PMD trained $\sim$40k steps.

### A.3. Details of the TFE Evaluation

The evaluation workflow begins with transcriptomic data from pre and post states, which can be at the bulk or single-cell level. Our conditional generative model TFE learns and encodes the gene expression changes caused by the disease, subsequently generating a perturbation representation **z**.

The perturbation representation **z**, carrying specific therapeutic knowledge, is then used as a query. We designed a cascaded filter with top-k for rapidly identifying structurally similar analogs of the perturbation representation in a large compound library. The core of this filter is a pre-built molecular fingerprint database, where matches are found by performing a Top-K nearest neighbor search. The computational engine relies on the Tanimoto similarity coefficient to quantify the similarity between the query **z** and a database molecule's fingerprint vector ($\mathbf{f}_d$). Through this process, we can efficiently screen for a set of known compounds that are most structurally similar to the perturbation representation **z**, which are then considered potential drug candidates for the specific patient or disease state.

### A.4. Details of the Gene Inhibitor (Extended Analysis of Molecular Docking and Binding Affinity)

To assess whether the molecules generated by CURE exhibit physical binding potential consistent with transcriptomic evidence, we moved beyond chemical similarity metrics to physics-based simulations. We employed molecular docking to calculate the binding affinity between the generated candidates and the crystal structures of the target proteins. This section details the quantitative results across the benchmark set and provides a qualitative visual analysis of the docking poses.

**Quantitative Analysis: Surpassing Baselines in Physical Binding.** As presented in Table 2, CURE consistently achieves the lowest binding energy (indicating the strongest affinity) across all 10 target genes. Notably, in targets such as EGFR and PIK3CA, our model achieves affinities significantly superior to the baseline methods. This quantitative advantage is non-trivial. The baseline models (e.g., TRIOMPHE, GexMolGen) often struggle to generate valid pharmacophores that fit tightly into specific protein pockets, resulting in higher docking energies (weaker binding). In contrast, CURE, driven by the heterogeneity-aware aggregation module (TFE-H), appears to capture the precise structural constraints required to induce

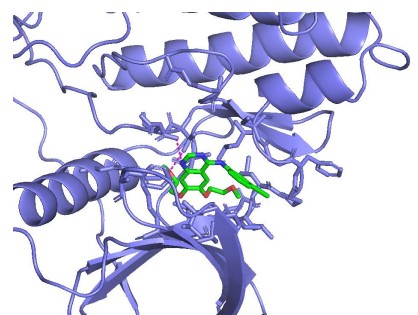

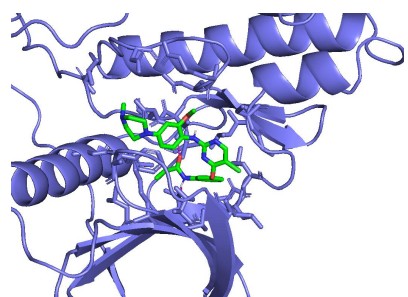

Ground Truth Molecule (Erlotinib) for EGFR
Affinity ↓ : -7.302

Generated Molecule for EGFR
Affinity ↓ : -9.164

*Figure 4.* Molecular docking validation against the EGFR kinase domain. The binding affinity of the CURE-generated candidate was evaluated using AutoDock Vina compared to the inhibitor Erlotinib (Ground Truth). The generated molecule which was random sampled by CURE exhibited a binding affinity of -9.164 kcal/mol, surpassing the reference Erlotinib (-7.302 kcal/mol) under identical simulation conditions. This indicates a stronger thermodynamic stability within the binding pocket.

the target phenotype, which inherently translates to high-affinity binding structures.

**Qualitative Case Study: Visualizing the EGFR Pocket.** To intuitively understand these results, we visualized the docking poses for the Epidermal Growth Factor Receptor (EGFR) target. Figure 4 compares the ground-truth inhibitor, Erlotinib, against a molecule generated by CURE under the zero-shot setting.

Ground Truth (Left): The known inhibitor Erlotinib docks into the ATP-binding pocket of EGFR with an affinity of $-7.302$ kcal/mol. Its quinazoline scaffold is well-positioned to interact with the hinge region residues.

Generated Molecule (Right): The molecule random sampled by CURE successfully occupies the same orthosteric binding site. Remarkably, it achieves an even stronger binding affinity of $-9.164$ kcal/mol. Visual inspection reveals that the generated molecule possesses a spatial configuration highly compatible with the pocket's geometry. It adopts a conformation that maximizes contact with the hydrophobic back pocket while positioning polar groups to potentially form stabilizing hydrogen bonds.

Crucially, while the generated molecule differs in exact chemical composition (SMILES) from Erlotinib, it preserves the essential 3D topology required for inhibition. This confirms that our model has performed a successful scaffold hopping, identifying a novel chemical entity that fulfills the same functional role.

**Bridging TBDD and SBDD: Function Informing Structure.** A notable implication of these results lies in the relationship between TBDD and SBDD. Conventionally, achieving high docking scores is the primary objective of SBDD, which explicitly conditions generation on the 3D geometry of the protein pocket. CURE, however, operates without ever seeing the protein structure; it is trained solely on transcriptomic perturbation data. The fact that CURE achieves docking scores comparable to, or even exceeding, those of known inhibitors suggests that the transcriptomic fingerprint of a perturbation may contain implicit information about the target's structural requirements. By learning to invert the gene expression signature, the model appears to capture pharmacophoric features required to trigger that signature. This suggests that function-oriented design can serve as a viable pathway to generate structurally competent ligands, even in scenarios where crystal structures are unavailable or difficult to obtain.

**A.5. Pseudocode for Training Loss**

---

**Algorithm 1** Pseudocode for View-1 Loss $\mathcal{L}_{\text{global}}$

---

1: **Input**: Graph matrix $\mathbf{X_G}$, TFE $F$, Transcription signals $\mathbf{T}_{\text{pre}}, \mathbf{T}_{\text{post}}$, VAE encoder $Q(z|\mathbf{X_G})$, VAE decoder $P(\mathbf{X_G}|z)$, KL weight $\lambda_{\text{KL}}$
2: **Output**: $\mathcal{L}_{\text{global}}$
3: $(\mu_{\text{enc}}, \sigma_{\text{enc}}) \leftarrow z_{\text{enc}} \leftarrow Q(\mathbf{X_G})$
4: $(\mu_f, \sigma_f) \leftarrow z_f \leftarrow F(\mathbf{T}_{\text{pre}}, \mathbf{T}_{\text{post}})$
5: $\mathcal{L}_{\text{ELBO}} \leftarrow -\mathbb{E}_{z_{\text{enc}} \sim Q}[\log P(\mathbf{X_G}|z_{\text{enc}})] + \lambda_{KL} D_{KL}[Q(z_{\text{enc}}|\mathbf{X_G})||P(z_{\text{enc}})]$       ▷ Standard ELBO
6: $\mathcal{L}_{\text{align}} \leftarrow ||\mu_{\text{enc}} - \mu_f||^2 + ||\sigma_{\text{enc}}^2 - \sigma_f^2||^2$
7: $\mathcal{L}_{\text{global}} \leftarrow \mathcal{L}_{\text{ELBO}} + \mathcal{L}_{\text{align}}$
8: **return** $\mathcal{L}_{\text{global}}$

---

**Algorithm 2** Pseudocode for View-2 Loss $\mathcal{L}_{\text{local}}$

---

1: **Input**: Predict vector $\mathbf{A}$, target fingerprint $\mathbf{B}$, SMILES label list $\mathbf{S}$, temperature $\tau = 0.1$, sparse weight $\lambda = 0.15$, $\alpha = 0.4$
2: **Output**: $\mathcal{L}_{\text{local}}$
3: $\mathcal{L}_{\text{sparse}} \leftarrow \text{RegressionLoss}(\mathbf{A}, \mathbf{B}, \alpha)$
4: $\mathcal{L}_{\text{InfoNCE}} \leftarrow \text{ContrastLoss}(\mathbf{A}, \mathbf{B}, \mathbf{S}, \tau, \lambda)$
5: $\mathcal{L}_{\text{local}} \leftarrow \mathcal{L}_{\text{sparse}} + \mathcal{L}_{\text{InfoNCE}}$
6: **return** $\mathcal{L}_{\text{local}}$
7: **function** CONTRASTLOSS($\mathbf{A}, \mathbf{B}, \mathbf{S}, \tau, \lambda$)
8:     **Input**: $\mathbf{A} \in \mathbb{R}^{b \times 2048}, \mathbf{B} \in \mathbb{R}^{b \times 2048}$ (non-negative integers), $\mathbf{S}$ (length $b$), $\tau, \lambda$
9:     **Output**: $\mathcal{L}_{\text{InfoNCE}}$
10:     $\mathbf{A} \leftarrow \text{Normalize}(\mathbf{A}), \mathbf{B} \leftarrow \text{Normalize}(\mathbf{B})$
11:     $\mathbf{Matrix} \leftarrow \mathbf{A} \cdot \mathbf{B}^{\top}/\tau$       ▷ Similarity matrix $\in \mathbb{R}^{b \times b}$
12:     $\mathbf{L} \leftarrow [0, 1, \ldots, b-1]$       ▷ Labels for diagonal elements
13:     $\mathbf{Mask}_{\text{same}} \leftarrow \text{Boolean}(S_i = S_j \text{ for all } i, j)$       ▷ Mask for same string labels
14:     $\mathbf{Mask}_{\text{same}}[\text{diagonal}] \leftarrow \text{False}$       ▷ Exclude diagonal
15:     $\mathbf{Matrix}[\mathbf{Mask}_{\text{same}}] \leftarrow -\infty$       ▷ Set non-diagonal same-label entries to large negative
16:     $\mathcal{L}_{\text{InfoNCE'}} \leftarrow \text{CrossEntropy}(\mathbf{Matrix}, \mathbf{L})$
17:     **if** $\lambda > 0$ **then**
18:         $\mathbf{Mask}_{\text{zero}} \leftarrow (\mathbf{B} = 0)$       ▷ Mask for zero positions in $\mathbf{B}$
19:         $\mathcal{L}_{\text{spa}} \leftarrow \text{Mean}((\mathbf{A} \cdot \mathbf{Mask}_{\text{zero}})^2)$
20:         $\mathcal{L}_{\text{InfoNCE}} \leftarrow \mathcal{L}_{\text{InfoNCE'}} + \lambda \cdot \mathcal{L}_{\text{spa}}$
21:     **end if**
22:     **return** $\mathcal{L}_{\text{InfoNCE}}$
23: **end function**
24: **function** REGRESSIONLOSS($\mathbf{A}, \mathbf{B}, \alpha$)
25:     **Input**: $\mathbf{A} \in \mathbb{R}^{b \times 2048}, \mathbf{B} \in \mathbb{R}^{b \times 2048}$ (non-negative integers), $\alpha$
26:     **Output**: $\mathcal{L}_{\text{sparse}}$
27:     $\mathbf{W} \leftarrow \text{ZerosLike}(\mathbf{B})$       ▷ Initialize weight matrix
28:     $\mathbf{Mask}_{\text{pos}} \leftarrow (\mathbf{B} > 0)$       ▷ Non-zero position mask
29:     $\mathbf{Mask}_{\text{neg}} \leftarrow (\mathbf{B} = 0)$       ▷ Zero position mask
30:     $\mathbf{W}[\mathbf{Mask}_{\text{pos}}] \leftarrow \log(1 + \mathbf{B}[\mathbf{Mask}_{\text{pos}}])$       ▷ Weights for non-zero positions
31:     $\mathcal{L}_{\text{pos}} \leftarrow \text{Sum}(\mathbf{W} \cdot (\mathbf{A} - \mathbf{B})^2 \cdot \mathbf{Mask}_{\text{pos}})/(\text{Sum}(\mathbf{Mask}_{\text{pos}}) + \epsilon)$
32:     $\mathcal{L}_{\text{neg}} \leftarrow \text{Sum}(\mathbf{A}^2 \cdot \mathbf{Mask}_{\text{neg}})/\text{Sum}(\mathbf{Mask}_{\text{neg}} + \epsilon)$
33:     $\mathcal{L}_{\text{sparse}} \leftarrow \mathcal{L}_{\text{pos}} + \alpha \cdot \mathcal{L}_{\text{neg}}$
34:     **return** $\mathcal{L}_{\text{sparse}}$
35: **end function**

---

## B. Data Construction and Splitting Protocols

In this section, we provide a granular description of the data processing pipelines and the rigorous splitting strategies employed to construct the evaluation benchmarks described in Section 5. All preprocessing steps utilized Python libraries `RDKit` (for molecular informatics) and `Scanpy` (for transcriptomic data).

### B.1. Dataset Preprocessing

**Bulk Cell Data (L1000).** We sourced the Level 3 profiles from the LINCS L1000 project (Subramanian et al., 2017). To ensure data quality, we filtered out experimental instances with low transcriptional consistency scores (distinct specificity $< 0.8$). **Single-Cell Data (Tahoe-100M).** For the Tahoe-100M dataset (Zhang et al., 2025), we performed standard quality control: cells with mitochondrial gene percentage $> 5\%$ or total gene counts $< 500$ were excluded. Raw counts were normalized by library size and log-transformed.

### B.2. Implementation of OOD Splitting Strategies

To systematically evaluate generalization, we curated three distinct dataset configurations. For each configuration, the final training/validation/testing ratio was maintained at approximately 85:10:5, but the *unit of splitting* differed fundamentally.

**1) Random Split (IID Setting).**  All drug-cell response pairs were pooled and randomly shuffled. This setting assumes Independent and Identically Distributed (IID) data and serves as a baseline to measure the model's capacity for interpolation within the training distribution.

**2) Unseen Drugs: The Scaffold-Split Protocol.**  To strictly enforce the **Scaffold-level Split** described in the main text, we employed the Bemis-Murcko scaffold decomposition algorithm provided by `RDKit`. The procedure is as follows:

1. **Scaffold Extraction:** For every unique drug in the dataset (L1000 and Tahoe-100M), we extracted its molecular scaffold by removing side chains and keeping the ring systems and linkers.

2. **Cluster Grouping:** Drugs sharing the exact same Bemis-Murcko scaffold were grouped into clusters.

3. **Stratified Partitioning:** Instead of splitting individual drugs, we split the *scaffold clusters*. This ensures that if a scaffold is assigned to the test set, *none* of the drugs sharing that scaffold appear in the training set.

4. **Filtering:** Trivial scaffolds (e.g., single benzene rings) containing an overwhelming number of drugs were downsampled in the training set to prevent class imbalance, while ensuring the test set contains complex, structurally distinct scaffolds.

This process guarantees that the test set requires the model to generalize to new chemical spaces rather than recalling neighbors from the training set.

**3) Unseen Cell Lines: The Hierarchical Biological Split Protocol.**  To implement the **Unseen Cell Lines** setting, we built a hierarchy of `Tissue → Tumor Type → Cell Line`.

1. **Hierarchy Construction:** Each cell line was mapped to its primary tissue of origin (e.g., Lung, Kidney, Breast) and specific disease subtype.

2. **Disjoint Separation:** We held out entire tissue groups or distinct tumor types for the test set. For instance, if "Lung Tissue" is selected for testing, all cell lines derived from lung tissue are strictly excluded from the training set.

## C. More Experimental Results and Discussions

### C.1. Impact of CFG Guidance Strength

We explored the effect of different CFG guidance strengths on the generation results (Table 6). We found that as the guidance strength increases, the performance metrics for molecular generation first rise and then slightly decline. This reveals a trade-off between potency and drug-likeness, providing a basis for selecting the optimal hyperparameter in practical

*Table 6.* Performance of training w/o cfg under different guidance scale. ↑ means higher is better, ↓ means lower is better.

| Metric | train w/o cfg | 0 | 1 | 2 | 3 | 4 | 5 | 6 | 7 | 8 | 9 |
|---|---|---|---|---|---|---|---|---|---|---|---|
| Validity↑ | 0.1934 | 0.0550 | 0.7025 | 0.7250 | **0.7500** | 0.6950 | 0.6950 | 0.6800 | 0.6750 | 0.6750 | 0.6300 |
| Coverage↑ | 63.64% | 54.55% | 72.73% | 72.73% | **90.91%** | 81.82% | 81.82% | 81.82% | 72.73% | 72.73% | 54.55% |
| Unique↑ | 0.7942 | 0.7222 | 0.7946 | 0.8007 | 0.8013 | 0.7984 | **0.8027** | 0.7998 | 0.8009 | 0.8011 | 0.8003 |
| Similarity↑ | 0.8691 | 0.7664 | **0.9598** | 0.9589 | 0.9581 | 0.9571 | 0.9573 | 0.9564 | 0.9573 | 0.9564 | 0.9574 |
| Distance↓ | 28.9203 | 45.4423 | 10.5358 | **9.3103** | 9.2467 | 10.4150 | 9.8974 | 10.2288 | 9.9570 | 10.0778 | 10.7401 |
| Fraggle Sim.↑ | 0.3441 | 0.3289 | 0.8734 | 0.8708 | **0.8896** | 0.8785 | 0.8705 | 0.8843 | 0.8800 | 0.8654 | 0.8600 |
| Morgan Sim.↑ | 0.2090 | 0.1281 | 0.8036 | 0.8076 | **0.8279** | 0.8187 | 0.8083 | 0.8239 | 0.8024 | 0.7906 | 0.8029 |
| QED↑ | 0.4952 | 0.4435 | 0.5435 | 0.5629 | **0.5673** | 0.5635 | 0.5650 | 0.5664 | 0.5671 | 0.5672 | 0.5632 |

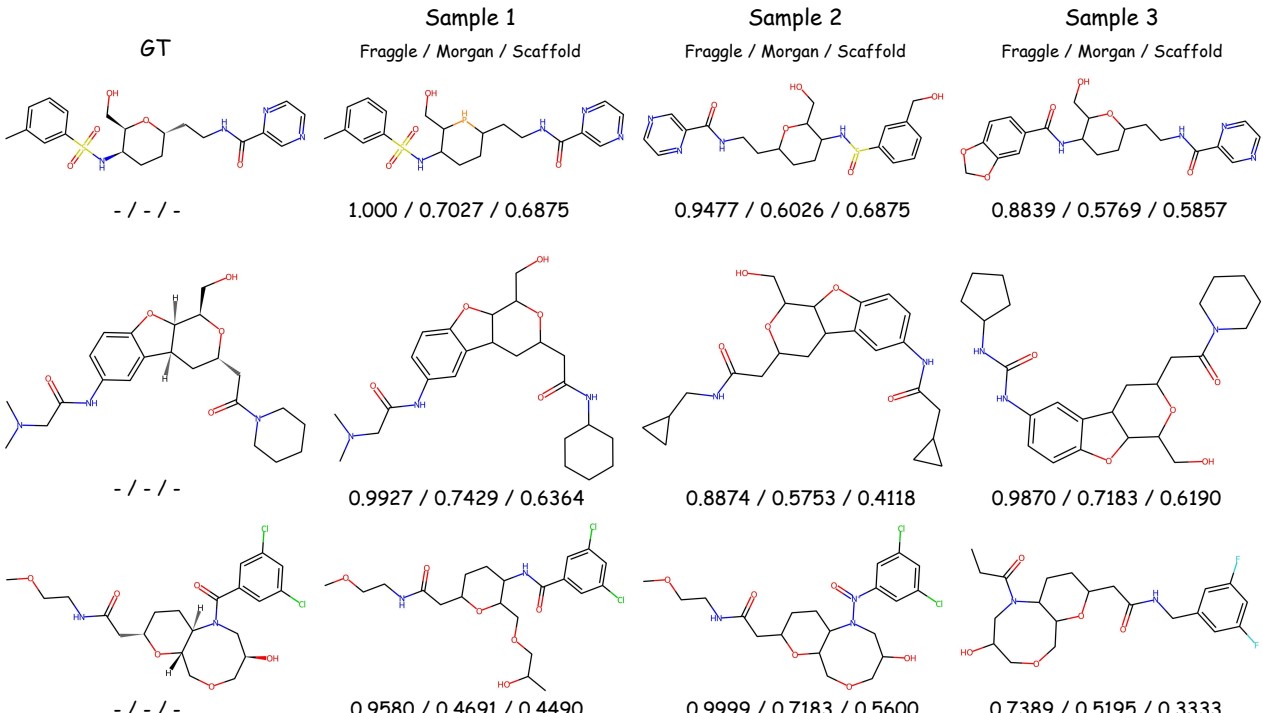

*Figure 5.* Molecular structure diagrams generated through multiple sampling. *\*Note: The indicators in the chart represent, from left to right: Fraggle/Morgan/Scaffold Sim. scores.*

applications. The experiment establishes a strength of 3 as the global optimum. The CFG strength acts as a lever to control potency and drug-likeness, low strength leads to insufficient potency, while high strength causes structural distortion. This finding provides a general theoretical framework for hyperparameter optimization in drug generation tasks. Unlike the previous experiments, which were tested on the entire dataset, this data was tested on a single batch (batchsize=200) to show the trend of the metrics.

## C.2. Molecular Structure Visualization

To further confirm that the high similarity does not result from memorization, we randomly sampled real drugs from the test set and compared them with model-generated candidates (Figure 5, Table 7). Crucially, while fingerprint-based metrics (Fraggle/Morgan) exhibited high similarity, the **Scaffold Similarity** (based on Bemis-Murcko decomposition) was notably lower. This divergence indicates that the generated molecules possess distinct chemical backbones despite sharing key functional properties. Consequently, the high fingerprint similarity stems from the preservation of pharmacophores rather than structural replication, demonstrating the model's capacity for *scaffold hopping* and effective function extraction.

*Table 7.* Molecular SMILE expressions and similarities generated from multiple sampling

| Target SMILES | Generated SMILES | Fraggle ↑ | Morgan ↑ | Scaffold ↓ |
|---|---|---|---|---|
| Cc1cccc(c1)S(=O)(=O)N[C@@H]1CC[C@@H](CCNC(=O)c2cnccn2)O[C@@H]1CO | Cc1cccc(S(=O)(=O)NC2CCC(CCNC(=O)c3cnccn3)PC2CO)c1 | 1.000 | 0.7027 | 0.6875 |
| | O=C(NCCC1CCC(NS(=O)c2cccc(CO)c2)C(CO)O1)c1cnccn1 | 0.9477 | 0.6026 | 0.6875 |
| | O=C(NC1CCC(CCNC(=O)c2cnccn2)OC1CO)c1ccc2c(c1)OCO2 | 0.8839 | 0.5769 | 0.5857 |
| CN(C)CC(=O)Nc1ccc2O[C@@H]3[C@@H](C[C@@H](CC(=O)N4CCCCC4)O[C@@H]3CO)c2c1 | CN(C)CC(=O)Nc1ccc2c(c1)C1CC(CC(=O)NC3CCCCC3)OC(CO)C1O2 | 0.9927 | 0.7429 | 0.6364 |
| | O=C(CC1CC2c3cc(NC(=O)CC4CC4)ccc3OC2C(CO)O1)NCC1CC1 | 0.8874 | 0.5753 | 0.4118 |
| | O=C(Nc1ccc2c(c1)C1CC(CC(=O)N3CCCCC3)OC(CO)C1O2)NC1CCCC1 | 0.9870 | 0.7183 | 0.6190 |
| COCCNC(=O)C[C@@H]1CC[C@@H]2[C@H](COC[C@H](O)CN2C(=O)c2cc(Cl)cc(Cl)c2)O1 | COCCNC(=O)CC1CCC(NC(=O)c2cc(Cl)cc(Cl)c2)C(COCC(C)O)O1 | 0.9580 | 0.4691 | 0.4490 |
| | COCCNC(=O)CC1CCC2C(COCC(O)CN2[N+](=O)c2cc(Cl)cc(Cl)c2)O1 | 0.9999 | 0.7183 | 0.5600 |
| | CCC(=O)N1CC(O)COCC2OC(CC(=O)NCc3cc(F)cc(F)c3)CCC21 | 0.7389 | 0.5195 | 0.3333 |

## C.3. Latent Space Analysis and Visualization of biological interpretability

Since CURE operates within the framework of TBDD, it lacks explicit indicators for target binding affinity. To rigorously investigate the biological interpretability of the model and verify whether the learned latent representations capture authentic mechanistic principles rather than mere statistical artifacts, we conducted a stratified visualization analysis using Uniform Manifold Approximation and Projection (UMAP) on the human gene inhibitor dataset. Our visualization strategy was designed to probe the latent space from two complementary perspectives: functional specificity and biological context sensitivity.

First, to validate the model's capability to encode mechanism-specific functional signatures, we isolated the cellular background by projecting latent embeddings of distinct gene inhibitors within a single cell line (U251MG, HT29, A549). As illustrated in the top row of Figure 6, the resulting manifold reveals a striking structural organization where samples form discrete, tight clusters according to the inhibitor type. This distinct separation implies that the model effectively extracts and encodes the unique transcriptomic perturbations associated with specific therapeutic targets, effectively mapping phenotypic changes to their underlying MoA.

Second, to demonstrate that the model maintains sensitivity to cellular heterogeneity and is not overfitting to a generic drug signature, we visualized the embeddings of identical inhibitors (MTOR, CTSK, SMAD3) across diverse cell lines. The bottom row of Figure 6 exhibits clear stratification driven by cellular identity, confirming that the model dynamically adapts its functional representations based on the biological context.

Collectively, these visualization results provide strong evidence for the model's validity. The ability to simultaneously achieve high intra-class compactness for inhibitors (demonstrating mechanistic understanding) and inter-class separability for cell lines (demonstrating context awareness) strongly suggests that the intermediate latent space operates as a biologically meaningful manifold. This indicates that our framework effectively disentangles the specific functional impact of drug perturbations from complex cellular background effects, establishing a robust foundation for function-oriented drug discovery.

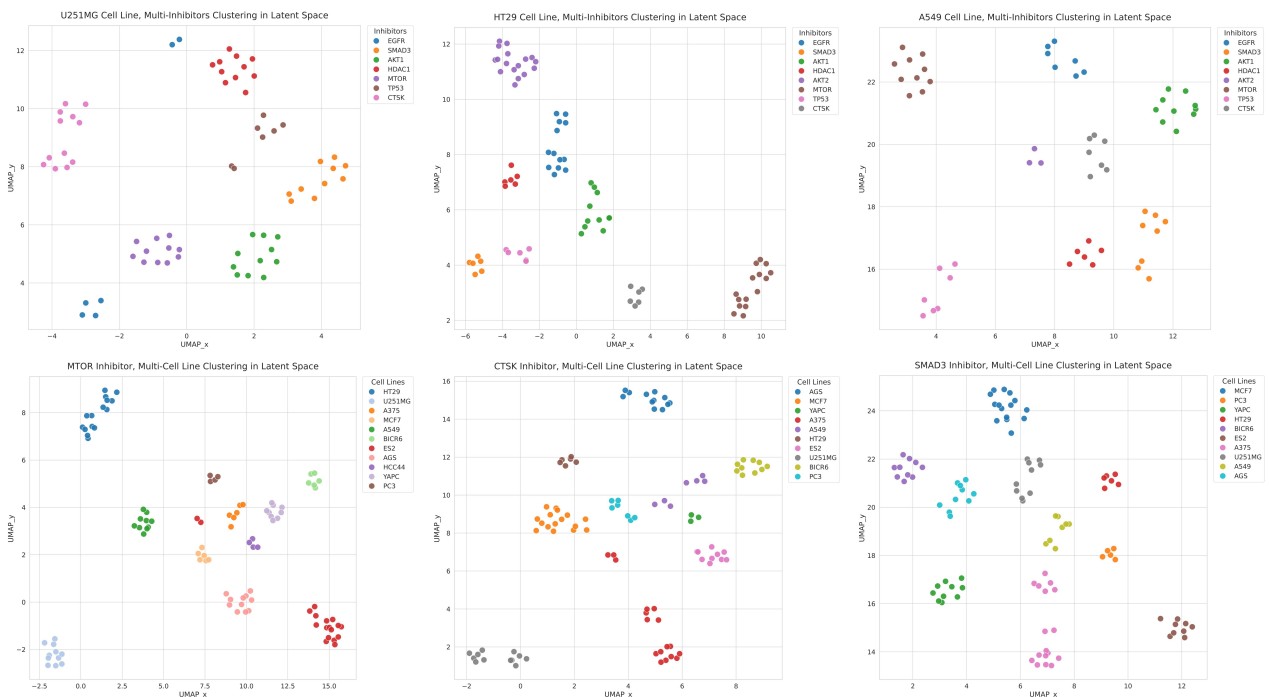

*Figure 6.* Latent space visualization using the human gene inhibitor dataset. UMAP projections reveal the model's ability to disentangle biological mechanisms from cellular contexts. (Top) Distinct clustering of different inhibitors within the same cell line demonstrates the encoding of mechanism-specific functional signatures. (Bottom) Stratification of identical inhibitors across diverse cell lines confirms the model's sensitivity to cellular heterogeneity.

## C.4. Evaluation of Toxicity Properties

To further assess the pharmacological viability and safety profile of the generated compounds, we extended our evaluation to include toxicity-related properties using the ADMETlab predictor (Fu et al., 2024). We compared molecules generated by CURE against those from baseline methods (GexMolGen, Gx2Mol, TRIOMPHE) as well as the ground-truth drugs from the L1000 test set. The evaluation covers a broad spectrum of toxicity risks, including mutagenicity (Ames), cardiotoxicity (hERG), and organ-specific toxicities. The results, summarized in Table 8, demonstrate that CURE achieves a competitive safety profile. Although our model was not explicitly optimized for these specific toxicity during training, the generated molecules exhibit toxicity scores that are consistently within a reasonable range, often matching or outperforming both the baseline methods and the ground-truth reference drugs (e.g., in Eye Irritation and Rat Oral Acute Toxicity).

*Table 8.* Comparison of predicted toxicity properties across generative models and the ground truth (L1000). Arrows indicate whether lower (↓) or higher (↑) scores are desirable.

| Metric | GexMolGen | Gx2Mol | TRIOMPHE | CURE | L1000 (GT) |
|---|---|---|---|---|---|
| Ames Mutagenicity ↓ | 0.5003 | **0.4632** | 0.5965 | 0.4850 | 0.5527 |
| hERG Blockers ($10\mu$M) ↓ | 0.4361 | 0.4236 | 0.4003 | **0.3983** | 0.2973 |
| Hematotoxicity ↓ | 0.4435 | **0.3436** | 0.3993 | 0.3590 | 0.5149 |
| Respiratory Toxicity ↓ | 0.4755 | 0.5332 | 0.8141 | **0.4684** | 0.4715 |
| Carcinogenicity ↓ | 0.4523 | **0.4511** | 0.5439 | 0.4714 | 0.5345 |
| DILI (Liver Injury) ↓ | 0.6968 | 0.6923 | 0.6623 | **0.6506** | 0.6733 |
| ROA (Rat Oral Acute Tox.) ↓ | 0.3475 | 0.3331 | 0.6658 | **0.3214** | 0.3414 |
| FDAMDD (Max Daily Dose) ↑ | 0.4875 | 0.5498 | **0.6136** | 0.5918 | 0.5272 |
| Eye Irritation ↓ | 0.1983 | 0.2082 | 0.2682 | **0.0942** | 0.2238 |
| Eye Corrosion ↓ | 0.0311 | 0.0264 | 0.1485 | **0.0141** | 0.0124 |

## C.5. Scaffold Novelty Analysis

To address concerns regarding whether high similarity metrics reflect structural duplication rather than genuine pharmacophoric capture, we performed a systematic scaffold novelty analysis on the bulk in-distribution setting. We report three complementary metrics: Unique Scaffold Ratio (fraction of unique Bemis-Murcko scaffolds among generated molecules), Scaffold Novelty (fraction of generated scaffolds absent from the training set), and Internal Diversity (mean pairwise Tanimoto distance among generated molecules).

*Table 9.* Scaffold novelty analysis on bulk in-distribution data.

| Method | Unique Scaffold Ratio ↑ | Scaffold Novelty ↑ | Internal Diversity ↑ |
|---|---|---|---|
| GexMolGen | 0.5373 | 0.6796 | 0.7646 |
| Gx2Mol | 0.4975 | 0.6337 | 0.8360 |
| TRIOMPHE | 0.5694 | 0.6637 | 0.8809 |
| **CURE** | **0.6360** | **0.7264** | **0.8906** |

As shown in Table 9, CURE achieves the highest scores across all three metrics. The high Scaffold Novelty (0.7264) confirms that over 72% of generated scaffolds are absent from training, while the co-occurrence of high fingerprint similarity (Table 1) and lower scaffold similarity (Figure 5) indicates successful scaffold hopping: preserving pharmacophores while exploring novel chemical backbones.

## C.6. Alternative Functional Proxy Validation

To verify that our evaluation conclusions are robust to the choice of functional proxy, we replaced PRnet with chemCPA (Hetzel et al., 2022), an independently validated perturbation predictor. As shown in Table 10, the method ranking is fully preserved under chemCPA, confirming that CURE's superiority in functional consistency is not an artifact of any particular evaluator.

*Table 10.* Functional proxy validation: PRnet vs. chemCPA. Method ranking is preserved.

| Method | PRnet MSE ↓ | chemCPA MSE ↓ |
|---|---|---|
| GexMolGen | 4.6504 | 5.0821 |
| Gx2Mol | 2.5987 | 2.9487 |
| TRIOMPHE | 7.4599 | 7.8536 |
| **CURE** | **0.2328** | **0.3415** |

## C.7. Single-Cell Baseline Robustness with Metacell Aggregation

No existing TBDD method natively handles single-cell input. Pseudo-bulk averaging is the standard adaptation strategy. To further strengthen the fairness of our comparison, we additionally tested metacell aggregation (Baran et al., 2019), which constructs representative cellular profiles via $k$-nearest-neighbor graph partitioning.

*Table 11.* Single-cell baseline robustness: pseudo-bulk vs. metacell aggregation (in-distribution).

| Aggregation | Method | Coverage ↑ | Morgan Sim. ↑ | PRnet MSE ↓ |
|---|---|---|---|---|
| Pseudo-bulk | GexMolGen | 54.55% | 0.2245 | 4.8549 |
| | Gx2Mol | 45.45% | 0.2884 | 4.1419 |
| | TRIOMPHE | 63.64% | 0.2011 | 7.7024 |
| Metacell | GexMolGen | 54.55% | 0.2492 | 4.3120 |
| | Gx2Mol | 54.55% | 0.3118 | 3.9315 |
| | TRIOMPHE | 63.64% | 0.2468 | 7.2836 |
| TFE-H | **CURE** | **90.90%** | **0.6114** | **0.4829** |

As shown in Table 11, metacell aggregation provides only marginal improvement over pseudo-bulk for baselines, while CURE maintains a substantial lead. The limitation is architectural: baseline methods accept a single aggregated vector and cannot model within-population heterogeneity, whereas TFE-H preserves sub-population structure.

### C.8. Generator Architecture Ablation

To isolate the contribution of the Graph Diffusion backbone from TFE conditioning, we replaced the generator while keeping TFE fixed.

*Table 12.* Generator architecture ablation with identical TFE conditioning (bulk in-distribution).

| Generator | Similarity ↑ | Coverage ↑ | Unique ↑ | Morgan Sim. ↑ | PRnet MSE ↓ |
|---|---|---|---|---|---|
| SMILES Decoder(Cheng et al., 2024) | 0.7406 | 72.73% | 0.6406 | 0.5192 | 4.6732 |
| Graph VAE(Jin et al., 2020) | 0.8176 | 72.73% | 0.7450 | 0.6845 | 2.5620 |
| **Graph Diffusion** | **0.9576** | **100.00%** | **0.8906** | **0.8228** | **0.2328** |

Graph Diffusion outperforms all alternatives across every metric (Table 12). The PRnet MSE gap (0.23 vs. 2.56 vs. 4.67) is especially striking, confirming that the graph diffusion backbone is critical for functional fidelity. Notably, even weaker generators produce condition-responsive molecules when guided by TFE, validating the effectiveness of our feature extraction.

### C.9. Extended TFE Input Processing Ablation

We compare different input processing strategies to demonstrate the co-dependence of TFE-I and TFE-A. The baseline uses officially pre-processed L1000 Level 5 data, while TFE variants operate on raw Level 3 with separate $T_{pre}$ and $T_{post}$.

*Table 13.* TFE input processing ablation (bulk in-distribution).

| Input Processing | Validity ↑ | Coverage ↑ | Diversity ↑ | Morgan Sim. ↑ |
|---|---|---|---|---|
| w/o TFE (Level 5) | 0.8775 | 90.91% | 0.7504 | 0.1824 |
| Concat($T_{pre}, T_{post}$) (Level 3) | 0.2150 | 36.36% | 0.7180 | 0.0752 |
| TFE-I only (Level 3) | 0.3000 | 63.64% | 0.7662 | 0.0886 |
| TFE-A only (Level 3) | 0.2400 | 36.36% | 0.6982 | 0.2527 |
| **Full TFE (Level 3)** | **0.9350** | **100.00%** | **0.8906** | **0.8228** |

As shown in Table 13, naive concatenation of Level 3 data severely degrades performance, and using TFE-I or TFE-A alone provides only partial recovery. Only the full TFE pipeline surpasses the Level 5 baseline by a large margin, confirming that TFE-I and TFE-A are co-dependent: TFE-I distills perturbation signals (without alignment, these remain noisy), while TFE-A aligns to chemical domains (without distillation, the alignment targets are incoherent).

### C.10. Docking Protocol and Controls

We provide the complete molecular docking protocol used for the zero-shot gene inhibitor evaluation (Section 5.4). All docking simulations were performed using AutoDock Vina with a search box of $25^3$ Å (except MTOR and SMAD3, which used $30^3$ Å due to larger binding sites).

As shown in Table 15, CURE consistently achieves binding affinities comparable to or exceeding those of known inhibitors across all 10 targets, demonstrating that transcriptomics-guided generation can produce physically viable binders.

## D. More Experimental Details

### D.1. Model Training Setup

We provide a comprehensive description of the model architecture complexity and the specific hyperparameter settings used during the training phases(Table 16, Table 17). For full reproducibility, we refer readers to the specific configuration files available in our source code repository.

*Table 14.* Docking protocol: PDB structures, grid centers, and known inhibitor controls.

| Target | PDB ID | Known Inhibitors | Grid Center (x, y, z) |
|--------|--------|------------------|-----------------------|
| AKT1 | 3O96 | MK-2206 | (8.37, −6.83, 12.62) |
| AKT2 | 2JDR | A-443654 | (21.81, 1.88, 42.42) |
| AURKB | 4C2V | Barasertib | (23.21, 0.30, 32.79) |
| CTSK | 1VSN | Odanacatib | (−2.72, 24.01, 6.33) |
| EGFR | 1M17 | Erlotinib | (22.01, 0.25, 52.79) |
| HDAC1 | 4BKX | Vorinostat | (−46.76, 16.29, −7.79) |
| MTOR | 4JT6 | Torkinib | (51.81, 0.00, −46.93) |
| PIK3CA | 7PG6 | Alpelisib | (−1.25, −9.01, 17.46) |
| SMAD3 | 1U7F | SIS3 | (−12.87, 36.04, 81.32) |
| TP53 | 2VUK | PhiKan083 | (124.68, 105.07, −43.12) |

*Table 15.* Binding affinity comparison (kcal/mol). Lower values indicate stronger binding.

| Target | GexMolGen ↓ | CURE ↓ | Known Inhib. ↓ |
|--------|-------------|--------|----------------|
| AKT1 | −7.45 | **−8.63** | −9.52 |
| AKT2 | −7.48 | **−8.59** | −8.14 |
| AURKB | −7.40 | **−8.79** | −10.20 |
| CTSK | −7.55 | **−8.69** | −8.58 |
| EGFR | −7.30 | **−9.11** | −7.33 |
| HDAC1 | −7.00 | **−8.68** | −8.48 |
| MTOR | −7.09 | **−8.74** | −8.22 |
| PIK3CA | −7.33 | **−9.15** | −8.74 |
| SMAD3 | −7.28 | **−9.07** | −8.17 |
| TP53 | −7.09 | **−8.28** | −7.69 |

## D.2. Details for PRnet

To rigorously quantify biological efficacy, we employed PRnet as a functional proxy: a flexible and scalable perturbation-conditioned generative model predicting transcriptional responses to novel complex perturbations at bulk and single-cell levels. We strictly enforced evaluation integrity by training the bulk PRnet model on the L1000 dataset (Subramanian et al., 2017; Gao et al., 2019) using the exact same training split as our generative framework, thereby eliminating any risk of data leakage from the test set. Furthermore, for the single-cell domain, the predictor was trained on the comprehensive Sci-plex dataset (Srivatsan et al., 2020), ensuring that our functional consistency metrics are derived from robust, domain-specific biological priors.

## D.3. Data Integrity and Prevention of Leakage

To ensure the validity of our evaluation and the generalization capability of the model, we strictly enforced data isolation protocols across all learnable modules. The TFE were trained exclusively on the designated training splits of the TBDD dataset, with no exposure to molecules or transcriptomes from the validation or test sets. Regarding the use of SCimilarity, it serves solely as a generic, frozen dimensionality-reduction tool. It was pre-trained on a broad human cell atlas for general cell-state embedding and was not fine-tuned on our L1000, Tahoe-100M, or ExCAPE datasets. Thus, it contains no task-specific supervision regarding drug-perturbation mappings. Similarly, the Morgan fingerprint alignment relies on deterministic RDKit computations without learning. These rigorous measures ensure that the model's performance stems from learning authentic structure-function mappings rather than data leakage or memorization.

*Table 16.* Hyperparameter Settings for Training Phases.

| Hyperparameter | TFE | PMD |
|---|---|---|
| Hardware | NVIDIA A100 (40GB) | NVIDIA A100 (40GB) |
| Total Training Time | $\sim 15$ GPU hours | $\sim 48$ GPU hours |
| Training Steps | 30k | 40k |
| Batch Size | 64 | 400 |
| Learning Rate | $1 \times 10^{-4}$ | $2 \times 10^{-4}$ |
| Optimizer | Adam | Adam |
| Diffusion Steps ($T$) | – | 500 |

*Table 17.* Summary of Model Parameters.

| Component | Description | Parameters |
|---|---|---|
| Diffusion Model | Graph Diffusion Transformer | $\sim 501.0$ M |
| TFE-I | Feature extraction and alignment modules | $\sim 7.8$ M |
| TFE-A | Encodes and reconstructs molecular graphs | $\sim 5.3$ M |

