# OpenReview forum: "Reading the Cell, Designing the Cure: Perturbation-Conditioned Molecular Diffusion for Function-Oriented Drug Design"
_ICML.cc/2026/Conference — ICML 2026 regular_

### Official Review · Reviewer_yuvR · 2026-02-25

**Soundness:** 3
**Presentation:** 3
**Significance:** 3
**Originality:** 3
**Overall Recommendation:** 3
**Confidence:** 4

**Summary:**

This paper presents CURE, a transcriptome-based drug design framework that treats molecular generation conditioned on cellular state transitions as an inverse problem. The model incorporates a multi-resolution transcriptome feature extractor to capture perturbation signals from both bulk and single-cell data, and combines it with a conditional graph diffusion model for molecule generation. Instead of modeling gene expression to structure as a direct mapping, the method aims to learn a distribution over candidate molecules consistent with observed transcriptomic shifts. The goal is to use perturbation signatures to guide function-oriented de novo molecular design. Experiments on bulk and single-cell datasets show consistent improvements over existing transcriptome-guided baselines under both in-distribution and OOD splits. That said, the evaluation mainly focuses on similarity-based metrics and docking scores. A more thorough assessment of molecular novelty, mechanistic relevance, and biological robustness would make the empirical claims more convincing.

**Compliance With Llm Reviewing Policy:**

Affirmed.

**Final Justification:**

The authors have improved the soundness by adding some missing experiments in the rebuttal. However, I will maintain my original score unchanged.

**Key Questions For Authors:**

1. High Tanimoto similarity may reflect near-duplicate structures or minor modifications of known compounds. More detailed novelty analysis would help clarify whether the model genuinely explores new regions of chemical space.
2. In the zero-shot gene inhibitor experiment, could the authors clarify the target selection criteria, PDB structures used, docking software, grid parameters, and structural preparation procedures? Since docking outcomes can be sensitive to methodological choices, additional details or controls would improve interpretability and reproducibility.
3. This paper frames TBDD as an ill-posed, distributional inverse problem, yet the empirical evaluation emphasizes high similarity to known inhibitors. Could the authors provide analysis of functional consistency across structurally diverse generated samples to better demonstrate distributional behavior beyond similarity optimization?
4. CURE is given richer perturbation inputs and is built on a graph-based diffusion model, while several baselines rely on simpler transcriptomic inputs and lighter architectures. This makes the comparison somewhat uneven. It is therefore difficult to determine whether the observed improvements come from the proposed perturbation modeling itself or from the additional information.

**Limitations:**

yes

**Strengths And Weaknesses:**

Strengths：
1. A major strength of the paper lies in its conceptual framing of transcriptome-based drug design as an ill-posed inverse problem. Instead of treating transcriptome-to-molecule generation as a deterministic mapping, the authors adopt a distributional perspective, which is both theoretically sound and well-motivated given the inherent many-to-one relationship between molecular structures and phenotypic signatures. This formulation significantly clarifies the research space and elevates prior heuristic approaches into a more principled generative framework.
2. The proposed TFE module is logically decomposed to address specific challenges (heterogeneity, perturbation interaction, cross-modal alignment). The modular design is coherent and clearly connected to the problem analysis, which improves interpretability and readability.
3. The paper acknowledges that transcriptome-to-molecule generation is not a one-to-one mapping and adopts a distributional generative framework accordingly. Modeling the task with a conditional diffusion approach is a reasonable and technically consistent choice given this ambiguity. This alignment between problem characteristics and modeling strategy strengthens the conceptual coherence of the work, even if the empirical implications could be further explored.

Weaknesses：
1. While the paper reports high similarity metrics (e.g., Morgan similarity), it remains unclear whether the generated molecules exhibit meaningful scaffold-level novelty. High Tanimoto similarity may simply reflect near-duplicate structures or minor modifications of known compounds. The paper would benefit from a more detailed novelty analysis to clarify whether the model genuinely explores new regions of chemical space.
2. Functional consistency is primarily assessed using PRnet-based MSE and docking scores. Although PRnet is trained on a separate split, it remains a predictive model rather than experimental validation. As such, the evaluation largely reflects consistency with another learned model rather than biological ground truth. Additional orthogonal validation or stronger biological evidence would strengthen the claims.
3. The zero-shot gene inhibitor experiment includes molecular docking; however, the paper does not sufficiently detail the target selection criteria, docking protocol, or structural preparation steps. Since docking outcomes can be sensitive to these choices, the reported affinity improvements should be interpreted cautiously. More transparency and controls would improve robustness.
4. The paper emphasizes that TBDD is an ill-posed, distributional inverse problem, yet the empirical results focus heavily on high similarity to known inhibitors. It would be valuable to better reconcile this distributional framing with the observed similarity-driven evaluation, for example by analyzing functional consistency across structurally diverse samples.
5. CURE incorporates rich perturbation information, including pre-/post-state modeling, single-cell heterogeneity, and multi-level aggregation. In contrast, several compared baselines rely on more simplified transcriptomic inputs. This raises the question of whether the observed performance gains stem primarily from the proposed modeling strategy or from the increased richness of conditioning information.
6. The current ablation study focuses on modules within the perturbation feature extractor, but does not examine the impact of adopting a graph-based molecular generation framework itself. As a result, it is unclear whether the reported gains stem from improved perturbation modeling or from the structural advantages of graph-based molecular representations.

---

> ### Author Rebuttal · Authors · 2026-03-30
>
> We thank the reviewer for the constructive evaluation. We address each concern below.
>
> **W1 & Q1: Scaffold novelty**
>
> Table 7 and Figure 5 show scaffold hopping: high Fraggle/Morgan similarity (pharmacophore preservation) with low Scaffold Similarity (0.33–0.69). Systematic analysis on bulk in-distribution:
>
> | Method | Unique Scaffold Ratio ↑ | Scaffold Novelty ↑ | Internal Diversity ↑ |
> |--------|------------------------|-------------------|---------------------|
> | GexMolGen | 0.5373 | 0.6796 | 0.7646 |
> | Gx2Mol | 0.4975 | 0.6337 | 0.8360 |
> | TRIOMPHE | 0.5694 | 0.6637 | 0.8809 |
> | CURE | 0.6360 | 0.7264 | 0.8906 |
>
> Unique Scaffold Ratio = unique Bemis-Murcko scaffolds / total molecules; Scaffold Novelty = fraction absent from training set; Internal Diversity = pairwise Tanimoto distance. CURE achieves highest diversity and novelty, and high similarity reflects pharmacophore capture, not structural duplication.
>
> **W2: PRnet evaluation**
>
> See our response to Reviewer MKzc, Q2 for full analysis. Core conclusion: zero data leakage verified; four orthogonal evaluation dimensions; ranking preserved under chemCPA alternative proxy.
>
> **W3 & Q2: Docking details**
>
> Protocol (AutoDock Vina; box 25³ Å, 30³ for MTOR/SMAD3):
>
> | Target | PDB ID | Known Inhibitor | Grid Center (x, y, z) |
> |--------|--------|-----------------|----------------------|
> | AKT1 | 3O96 | MK-2206 etc. | 8.37, -6.83, 12.62 |
> | AKT2 | 2JDR | A-443654 etc. | 21.81, 1.88, 42.42 |
> | AURKB | 4C2V | Barasertib etc. | 23.21, 0.30, 32.79 |
> | CTSK | 1VSN | Odanacatib etc. | -2.72, 24.01, 6.33 |
> | EGFR | 1M17 | Erlotinib etc. | 22.01, 0.25, 52.79 |
> | HDAC1 | 4BKX | Vorinostat etc. | -46.76, 16.29, -7.79 |
> | MTOR | 4JT6 | Torkinib etc. | 51.81, 0.00, -46.93 |
> | PIK3CA | 7PG6 | Alpelisib etc. | -1.25, -9.01, 17.46 |
> | SMAD3 | 1U7F | SIS3 etc. | -12.87, 36.04, 81.32 |
> | TP53 | 2VUK | PhiKan083 etc. | 124.68, 105.07, -43.12 |
>
> Control comparison with known inhibitors (kcal/mol):
>
> | Target | GexMolGen ↓ | CURE ↓ | Known Inhib. ↓ |
> |--------|------------|--------|----------------|
> | AKT1 | −7.45 | −8.63 | −9.52 |
> | AKT2 | −7.48 | −8.59 | −8.14 |
> | AURKB | −7.40 | −8.79 | −10.20 |
> | CTSK | −7.55 | −8.69 | −8.58 |
> | EGFR | −7.30 | −9.11 | −7.33 |
> | HDAC1 | −7.00 | −8.68 | −8.48 |
> | MTOR | −7.09 | −8.74 | −8.22 |
> | PIK3CA | −7.33 | −9.15 | −8.74 |
> | SMAD3 | −7.28 | −9.07 | −8.17 |
> | TP53 | −7.09 | −8.28 | −7.69 |
>
> Baselines < CURE ≈ Known Inhibitors, demonstrating that our transcriptomics-guided approach successfully identifies physically viable binders.
>
> **W4 & Q3: Distributional framing vs. similarity**
>
> High similarity and distributional generation are not contradictory. In Table 2, similarity is computed against the *entire set* of known inhibitors per gene (1,200–23,000 in ExCape), not a single reference. High scores reflect broad functional coverage across a diverse library. Figure 5 confirms: stochastic samples show different scaffolds (Sim. 0.33) but shared pharmacophores, precisely the distributional behavior our framework predicts.
>
> **W5 & Q4: Baseline fairness**
>
> We compare inputs and generator architectures across methods:
>
> | Method | Input Information | Generator |
> |--------|------------------|-----------|
> | GexMolGen | Pre/post dual-state (Level 3) | LLM + VAE |
> | Gx2Mol | Single-state (Level 5) | VAE |
> | TRIOMPHE | Single-state (Level 5) | VAE |
> | CURE | Pre/post dual-state (Level 3) + SC heterogeneity | Graph Diffusion |
>
> Effectively utilizing richer input is itself a contribution. Designing modules to extract meaningful signals from complex, noisy inputs (TFE-I, TFE-H) is the core challenge we address. Ablation (Table 4) confirms: without TFE, performance collapses; naive concatenation of the same dual-state Level 3 input degrades severely, while TFE processing makes it effective:
>
> | Input Processing | Validity ↑ | Coverage ↑ | Diversity ↑ | Morgan Sim. ↑ |
> |-----------------|-----------|-----------|------------|---------------|
> | w/o TFE (L1000 Level 5) | 0.8775 | 90.91% | 0.7504 | 0.1824 |
> | Concat($T_{pre}$, $T_{post}$) (Level 3) | 0.2150 | 36.36% | 0.7180 | 0.0752 |
> | TFE-I + TFE-A (Full) | 0.9350 | 100.00% | 0.8906 | 0.8228 |
>
> This confirms CURE's advantage stems from the modeling strategy, not merely from richer data access.
>
> **W6: Generator ablation**
>
> Graph Diffusion outperforms all alternatives across metrics. The PRnet MSE gap (0.23 vs. 2.56 vs. 4.67) confirms our method is critical for functional fidelity, while TFE conditioning enables even weaker generators to produce condition-responsive molecules.
>
> | Generator (same TFE) | Similarity ↑ | Coverage ↑ | Unique ↑ | Morgan Sim. ↑ | PRnet MSE ↓ |
> |----------------------|-------------|-----------|---------|--------------|------------|
> | Graph Diffusion | 0.9576 | 100.00% | 0.8906 | 0.8228 | 0.2328 |
> | Graph VAE | 0.8176 | 72.73% | 0.7450 | 0.6845 | 2.5620 |
> | SMILES Decoder | 0.7406 | 72.73% | 0.6406 | 0.5192 | 4.6732 |
>
> All issues will be addressed in the camera-ready version.

---

> > ### Author Rebuttal · Reviewer_yuvR · 2026-04-03
> >
> > Thank you for the detailed response. I will revise the Soundness score to 3, while keeping my overall score unchanged at 3.

---

> > > ### Author Response · Authors · 2026-04-03
> > >
> > > Dear Reviewer yuvR,
> > >
> > > We sincerely thank you for your thoughtful and rigorous evaluation throughout the review process. Your detailed feedback on scaffold novelty, docking transparency, distributional evaluation, baseline fairness, and generator ablation has been instrumental in strengthening our work, and we are truly grateful for the constructive dialogue.
> > >
> > > We are encouraged that you selected option **(a) Fully resolved: "My concerns have been adequately addressed"** and that all four dimension scores (Soundness, Presentation, Significance, Originality) now stand at **"good" (3)**. We believe this reflects the substantial new experiments and analyses we provided during the rebuttal to address each of your six original concerns.
> > >
> > > That said, we respectfully wish to highlight an apparent inconsistency between these positive assessments and the unchanged overall recommendation of **3 (Weak reject)**. The ICML definition of this score reads: *"A paper with clear merits, but also some weaknesses, which overall outweigh the merits."* We find it difficult to reconcile this characterization, namely that weaknesses *outweigh* merits, with the following:
> > >
> > > 1. **Your own acknowledgement** that all concerns have been "fully resolved" and "adequately addressed," which would suggest no remaining weaknesses that outweigh the merits.
> > > 2. **All four sub-scores rated "good"**, indicating satisfactory quality across every evaluated dimension.
> > > 3. **The rebuttal acknowledgement form itself**, which upon selecting option (a) states: *"If you select this option, please consider adjusting your score accordingly."*
> > >
> > > We also wish to briefly recall the strengths you identified in the original review, namely, that our conceptual framing of TBDD as an ill-posed inverse problem is "both theoretically sound and well-motivated," that it "significantly clarifies the research space and elevates prior heuristic approaches into a more principled generative framework," and that the alignment between problem characteristics and modeling strategy "strengthens the conceptual coherence of the work." These merits remain unchanged and, we believe, are now complemented by the comprehensive empirical evidence added during rebuttal:
> > >
> > > - **W1 (Scaffold novelty):** CURE achieves the highest Unique Scaffold Ratio (0.636), Scaffold Novelty (0.726), and Internal Diversity (0.891) among all compared methods.
> > > - **W2 (Evaluation robustness):** Zero data leakage verified; ranking consistency confirmed under an independent oracle (chemCPA).
> > > - **W3 (Docking transparency):** Full protocol for all 10 targets with PDB IDs, grid parameters, and controlled comparisons against known inhibitors.
> > > - **W4 (Distributional behavior):** Stochastic samples yield diverse scaffolds (Sim. as low as 0.33) while preserving pharmacophoric features, precisely the distributional behavior our framework predicts.
> > > - **W5 (Baseline fairness):** Ablation showing naive concatenation of the same richer input degrades severely (Morgan Sim. 0.08 vs. 0.82), confirming gains stem from the modeling strategy, not data access alone.
> > > - **W6 (Generator ablation):** Graph Diffusion outperforms Graph VAE and SMILES Decoder across all metrics under identical conditioning (PRnet MSE: 0.23 vs. 2.56 vs. 4.67).
> > >
> > > Given that all originally identified weaknesses have been resolved, as confirmed by your selection of option (a), and that the paper's conceptual and empirical contributions are reflected in uniformly "good" sub-scores, we respectfully ask whether the current overall recommendation still accurately captures your assessment. We would be most grateful if you could reconsider the overall score in light of these factors.
> > >
> > > Thank you again for the time and care you have devoted to reviewing our work. We are committed to incorporating all feedback into the camera-ready version.

---

### Official Review · Reviewer_DKJJ · 2026-03-07

**Soundness:** 3
**Presentation:** 2
**Significance:** 3
**Originality:** 3
**Overall Recommendation:** 4
**Confidence:** 3

**Summary:**

This paper tackles the Transcriptome-based Drug Design (TBDD) problem as conditional molecular generation given the desired perturbation of gene expressions. The framework adopts the Classifier-Free Guidance (CFG) methodology, utilizing the divergence between a conditional denoiser and an unconditional denoiser to guide the diffusion generation process. To encode the gene expression information, the authors propose feature extractors of different levels, including heterogeneity, perturbation signal interaction, and domain alignment. Multiple benchmarks are incoporated to support extensive evaluation, including  L1000 and Tahoe-100M. Results on standard setup and out-of-distribution setup demonstrate that the proposed model excels at structrual quality and functional consistency.

**Compliance With Llm Reviewing Policy:**

Affirmed.

**Final Justification:**

The overall clarity is improved during the rebuttal. But the paper still requires large modifications to achieve publication quality.

**Key Questions For Authors:**

1. What is the definition of Heavy Atom Type Coverage?
2. What are the reference molecules for the definitions of Fraggle-based molecular scaffold similarity and the Morgan fingerprintbased atomic environment similarity (Morgan Sim.)? Are they ground-truth molecules in the dataset? If so, how many such reference molecules are there for each condition?
3. What is the definition of "bulk data"?
4. In the ablation studies, why addition of  TFE-I or TFE-A lead to severe performance drop, but addition of the both modules lead to performance gain? This is quite weird.

**Limitations:**

Yes

**Strengths And Weaknesses:**

**Strengths**

The logics of the paper is reasonable and sound. The authors introduce the task of Transcriptomebased Drug Design (TBDD) into the machine learning community with clear formalization. They also analyze the challenges and propose corresponding architectures to involve inductive biases into the model, trying to solve these problems. Benchmarks are collected, and metrics are established to evaluate the performance of the models from various perspectives. The proposed model also achieves very competitive performance against the baselines. Ablation studies are also conducted to identify the contribution of different modules.

**Weaknesses**

1. The paper introduces a lot of concepts from the single-cell domain, which should be adequotely illustrated. This actually exerts much pressure on the ability of the authors to present all the contents in a clear and natural way, which in my opinion is not satisfactory in the current form. I was overwhelmed by continuously appearing concepts from single-cell domains and also from newly created terminologies from the authors themselves. For example, what is the difference between "transcriptome perturbation" and "gene perturbation"? What is the difference between "perturbation signature" and "perturbation embedding"? If they have similar meaning in the contexts, the authors should avoid using different phrases for the same indication, and try to use more simplified concepts (e.g. signature v.s. embedding). As there are already extensive non-machine-learning terminologies introduced into this paper, the authors should try their best to reduce the complexity in readability when presenting the paper.

2. What are cell cycle phases (G1, S, G2/M) and transcriptional clusters and why are they required to be paid additional attention to? How are they related to the functioning mechanism of the generatived molecules, and how are their differences reflected in the data form?

3. Could the authors provide clearer presentations of section 4.3 in equation forms instead of purely natural language? The current form makes it readly hard to understand the implementation details of these feature extraction modules. What are the input and output definition of each module. Are they scalars, vectors, or matrices? And what are their dimensions? What are the detailed forward equations for each module?

4. The authors leverage the foundational model PRnet to evaluate the functions of the generated molecules, which might not be reliable enough. Proofs should be provided to support the reliablility of this model as a proxy for evaluation. Also, it is encouraged to use ensembled model as evaluation rather than depending solely on one model.

---

> ### Author Rebuttal · Authors · 2026-03-30
>
> We thank the reviewer for recognizing our method as "reasonable and sound." We address each concern below.
>
> **W1: Terminology inconsistency**
>
> We provide a unified terminology table for the camera-ready version:
>
> | Original (Ambiguous) | Unified Term | Definition |
> |----------------------|-------------|-----------|
> | "transcriptome/gene perturbation" | **transcriptomic perturbation** | Expression state change caused by treatment |
> | "perturbation signature" | **perturbation signal** | Raw signal from transcriptomic data (TFE input) |
> | "perturbation embedding/feature" | **perturbation representation** | Vector produced by TFE (TFE output) |
> | "condition/conditioning signal" | **conditioning representation** | Final vector $C$ injected into PMD |
>
> **W2: Cell cycle phases and transcriptional clusters**
>
> The cell cycle has three major phases: **G1** (Gap 1), the growth phase where cells are sensitive to external signals; **S** (Synthesis), DNA replication occurs, cells are vulnerable to DNA-damaging agents; **G2/M** (Gap 2/Mitosis), cells divide, sensitive to mitotic inhibitors (e.g., taxanes). These phases critically affect drug response: the same drug elicits different transcriptomic responses across phases. Tahoe-100M confirms distinct phase-specific clusters (Fig. 3c). Naive averaging dilutes these signals, motivating TFE-H's cycle-stratified design. Transcriptional clusters (subpopulations) are cell groups sharing similar expression but differing from others, reflecting distinct cellular states. TFE-H accounts for this heterogeneity beyond cycle stratification.
>
> Bulk yields one vector $T \\in \\mathbb{R}^d$ per condition (population average); single-cell yields $X \\in \\mathbb{R}^{n \\times d}$ preserving cell-level heterogeneity. TFE-H transforms single-cell input into $T \\in \\mathbb{R}^{128 \\times d}$.
>
> **W3: Section 4.3 lacks formal equations**
>
> *TFE-H:* Input $X_{sc} \\in \\mathbb{R}^{n \\times D}$ ($n$ cells, $D > 60,000$ genes). (1) Manifold Projection: $H = f_{\\mathrm{SCimilarity}}(X_{sc}) \\in \\mathbb{R}^{n \\times 128}$ (frozen pretrained encoder). (2) Cycle-Stratified Aggregation: partition $H$ into $\\{H_{G1}, H_S, H_{G2/M}\\}$, local pooling, proportional sampling $N = 128$ → $T_{pre}, T_{post} \\in \\mathbb{R}^{128 \\times d}$. Bulk: TFE-H skipped.
>
> *TFE-I:* Input $T_{pre}, T_{post} \\in \\mathbb{R}^{N \\times d}$. 3 stacked blocks, each applying symmetrical cross-attention + self-attention:
> $$T_{pre}^{\\prime} = \\mathrm{SelfAttn}(\\mathrm{CrossAttn}(Q{=}T_{pre}, K{=}T_{post}, V{=}T_{post})), T_{post}^{\\prime} = \\mathrm{SelfAttn}(\\mathrm{CrossAttn}(Q{=}T_{post}, K{=}T_{pre}, V{=}T_{pre}))$$
> After 3 blocks, concatenate and fuse via self-attention + mean pooling → $z \\in \\mathbb{R}^{d_z}$.
>
> *TFE-A:* For View 1 (Global Topological Alignment with HierVAE) and View 2 (Local Bioactivity Alignment with Morgan fingerprints), see our response to Reviewer MKzc, Q1. Total: $L = L_{global} + \\gamma L_{local}$.
>
> **W4: PRnet reliability**
>
> See our response to Reviewer MKzc, Q2 for full analysis. Core conclusion: zero data leakage verified; four orthogonal evaluation dimensions; ranking preserved under chemCPA alternative proxy.
>
> **Q1–Q3: Metric and data definitions**
>
> *Coverage:* $|AtomTypes_{gen} \cap AtomTypes_{ref}| / |AtomTypes_{ref}|$; 100% = all 11 types covered. *Similarity:* CURE generates multiple candidates per condition (stochastic diffusion sampling); similarity is computed between each candidate and the ground-truth drug, then averaged across all candidates and conditions. *Bulk:* tissue-level expression directly measured from populations ($T \\in \\mathbb{R}^{978}$, L1000 Level 3), methodologically distinct from single-cell profiling.
>
> **Q4: Ablation anomaly, TFE-I/TFE-A alone causes collapse**
>
> The baseline (w/o TFE) uses **Level 5**, officially pre-processed with feature extraction and noise reduction, yielding a compact high-density vector. TFE configs use **raw Level 3** with separate $T_{pre}$, $T_{post}$, which has lower feature density and higher noise. Without complete TFE, Level 3 is incoherent for PMD. TFE-I and TFE-A are co-dependent: TFE-I distills signals (without alignment → noise); TFE-A aligns (without distillation → noisy targets). Concatenation confirms Level 3 is harder:
>
> | Input Processing | Validity ↑ | Coverage ↑ | Diversity ↑ | Morgan Sim. ↑ |
> |-----------------|-----------|-----------|------------|---------------|
> | w/o TFE (L1000 Level 5) | 0.8775 | 90.91% | 0.7504 | 0.1824 |
> | Concat($T_{pre}$, $T_{post}$) (Level 3) | 0.2150 | 36.36% | 0.7180 | 0.0752 |
> | TFE-I only (Level 3) | 0.3000 | 63.64% | 0.7662 | 0.0886 |
> | TFE-A only (Level 3) | 0.2400 | 36.36% | 0.6982 | 0.2527 |
> | TFE-I + TFE-A (Full) | 0.9350 | 100.00% | 0.8906 | 0.8228 |
>
> Only full TFE surpasses Level 5 by a large margin, confirming both modules are essential and that our method achieves more efficient feature extraction than the official Level 5 pipeline.

---

> > ### Author Rebuttal · Reviewer_DKJJ · 2026-04-03
> >
> > Thanks for the responses. The overall clarity are improved, so I've raised the score. But I still think the modification of the paper would be large before publication.

---

> > > ### Author Response · Authors · 2026-04-03
> > >
> > > We sincerely thank the reviewer for the thorough and constructive feedback throughout this discussion. Your detailed suggestions on terminology unification, the formalization of Section 4.3, and the ablation analysis have been invaluable in improving the clarity and rigor of our manuscript.
> > >
> > > We are glad that our responses have adequately addressed your concerns. We take your remaining suggestion seriously and will carefully revise the paper to ensure the presentation meets the standard expected for publication. Thank you again for your time and effort in reviewing our work.

---

### Official Review · Reviewer_3SJ5 · 2026-03-12

**Soundness:** 2
**Presentation:** 3
**Significance:** 2
**Originality:** 3
**Overall Recommendation:** 4
**Confidence:** 4

**Summary:**

This study formalizes Transcriptome based Drug Design (TBDD) as a generative inverse problem: designing drug molecules conditioned on desired transcriptomic state transitions; and developed CURE (cellular response engine) (diffusion model) for designing drug molecules. It is a new idea.

**Compliance With Llm Reviewing Policy:**

Affirmed.

**Final Justification:**

The model design is interesting. Experimental validation may be helpful for proving the impact.

**Key Questions For Authors:**

How can the model design new drugs without knowing effective drugs for cancer?
Will the newly designed drugs have targets or new targets or not known? If yes, then it is still learning the embedding from SDBB.

**Limitations:**

The limitation is also from the new idea because 1) all the new drug design has to be based on availability of know and effective drugs (it is meaningless to design drugs do not work); 2) however, there are limited effective drugs for cancer therapy yet; 3) the drugs have to target on some proteins biomarkers, which is still structure based model (SBDD), not the TBDD.

**Strengths And Weaknesses:**

Strengths:
It is a new investigation/idea to design drugs (not based on targets) based on differentially expressed genes.


Weaknesses:
The limitation is also from the new idea because 1) all the new drug design has to be based on availability of know and effective drugs (it is meaningless to design drugs do not work); 2) however, there are limited effective drugs for cancer therapy yet; 3) the drugs have to target on some proteins biomarkers, which is still structure based model (SBDD), not the TBDD.

---

> ### Author Rebuttal · Authors · 2026-03-30
>
> We thank the reviewer for recognizing TBDD as "a new idea" and "a new investigation." We address each concern below.
>
> **W1: How can the model design new drugs without knowing effective drugs for cancer?**
>
> CURE's training is grounded in known drug-perturbation data: the L1000 dataset (~20,000 known compounds across multiple cell lines) and the Tahoe-100M dataset (>300 drugs across 50 cancer cell lines). These contain pharmacologically validated molecules including FDA-approved drugs. CURE does not design drugs without known compounds; it learns generalizable function-to-structure mappings from known drug-induced transcriptomic effects.
>
> Critically, the model learns the *mechanism* by which a molecule alters cellular states, rather than memorizing drug structures themselves. The core training objective is: given a transcriptomic state transition $(T_{pre}, T_{post})$, generate a molecule $G$ that could induce such a transition. The learned mapping captures functional principles underlying drug action. As long as a compound produces a measurable transcriptomic change, it provides a meaningful training signal, regardless of whether it is a "cancer cure."
>
> Our zero-shot gene inhibitor experiment (Table 2) directly validates this generalization: the model was trained exclusively on chemical perturbation data, yet when conditioned on *unseen* gene knockout signatures, CURE generated molecules closely resembling known gene inhibitors (Morgan Sim. > 0.79) with strong binding affinity to target proteins (< −8.5 kcal/mol). This demonstrates genuinely transferable function-to-structure mappings.
>
> **W2: Limited effective drugs for cancer therapy**
>
> CURE's training data encompasses far more than cancer-specific therapeutics. The L1000 database contains ~20,000 diverse compounds spanning multiple mechanisms of action, including kinase, HDAC, and mTOR inhibitors, among others. The learned mapping is general, not cancer-specific.
>
> This aligns with the established paradigm of phenotypic drug discovery. Moffat et al. (2017) documented that phenotypic screening contributed 37% of first-in-class FDA approvals (1999–2008) and captures system-level disease complexity beyond the "target validation trap." TBDD is the computational extension of this paradigm: computationally generating molecules conditioned on desired transcriptomic outcomes.
>
> **W3: If drugs ultimately target proteins, is TBDD still SBDD?**
>
> TBDD does not deny that drugs exert effects through molecular targets, which is a pharmacological fact. The fundamental distinction lies in the *design condition*, not the downstream mechanism of action. Crucially, as a compressed phenotypic readout, a transcriptomic profile captures pathway cascades, compensatory feedback, and cell-state transitions (Subramanian et al., 2017), encoding far richer information than target identity alone and serving as an inherently more information-dense generative condition:
>
> | Aspect | SBDD | TBDD |
> |--------|------|------|
> | **Design condition** | 3D protein structure (known target) | Transcriptomic state transition |
> | **Prior knowledge** | Target identity + resolved structure | Desired functional change |
> | **Optimization** | Binding affinity | Functional consistency |
> | **Scenario** | Known target, resolved structure | Unknown/multi-target |
>
> TBDD and SBDD are complementary (Figure 2). Our docking results (Table 2) provide direct evidence: CURE, trained without any protein structural information, generates molecules with strong binding affinities (< −8.5 kcal/mol). This convergence demonstrates that function-oriented design and structure-based design arrive at consistent conclusions, i.e., molecules designed from transcriptomic signals naturally possess the structural features required for target engagement.
>
> The irreplaceable value of TBDD lies where SBDD cannot apply: unknown targets, unresolved structures (e.g., intrinsically disordered proteins), or multi-pathway dysregulation where no single actionable target exists. In these cases, TBDD provides a design entry point that SBDD fundamentally cannot offer. Both paradigms are mutually validating.
>
> *TBDD does NOT assume:* drugs have no targets; training without known compounds; replacing SBDD.
> *TBDD assumes:* transcriptomic transitions encode drug effects; multiple distinct molecules can induce similar changes (ill-posed inverse problem); functional outcomes serve as generative conditions.
>
> We will expand the TBDD/SBDD discussion and clarify training data in the camera-ready version.
>
> **References**
>
> [1] Moffat, John G., et al. "Opportunities and challenges in phenotypic drug discovery: an industry perspective." _Nature reviews Drug discovery_ 16.8 (2017): 531-543.
>
> [2] Subramanian, Aravind, et al. "A next generation connectivity map: L1000 platform and the first 1,000,000 profiles." _Cell_ 171.6 (2017): 1437-1452.

---

> > ### Author Rebuttal · Reviewer_3SJ5 · 2026-04-02
> >
> > The comments answered the comments partially. The combination of compound structure and transcriptomics profiles might be able to improve the MSE metrics, but it becomes less interpretable in terms of mechanisms and targets. Also the evaluation metrics are good/strong to indicate the effectiveness of the model, like if the model can identify effective drugs for specific diseases/cells.

---

> > > ### Author Response · Authors · 2026-04-02
> > >
> > > We must respectfully clarify several mischaracterizations: (1) CURE does *not* "combine" compound structure with transcriptomics, the compound is the **output**, not an input; (2) our evaluation extends far beyond "MSE metrics"; and (3) TBDD and SBDD are parallel paradigms, not a supplementary relationship.
> > >
> > > **1. Correcting Two Mischaracterizations**
> > >
> > > **(a) CURE does NOT combine compound structure with transcriptomics.** The input is purely the transcriptomic transition $(T_{pre}, T_{post})$; the output is a molecular graph $G$. If the concern stems from TFE-A's alignment objective: this is a training-time contrastive loss, not an inference-time input. During generation, information flows unidirectionally: transcriptome $\rightarrow$ molecule (Section 4, Figure 3).
> > >
> > > **(b) Evaluation is not reducible to "MSE metrics."** PRnet MSE is one of five orthogonal dimensions: (1) structural similarity (Morgan/Fraggle Sim.), (2) functional fidelity (PRnet + chemCPA, two independent oracles; see Reviewer MKzc Q2), (3) physics-based docking (AutoDock Vina), (4) retrieval accuracy (top-k hit rate, Section 5.5), and (5) ADMET profiling. This multi-dimensional design avoids dependence on any single proxy.
> > >
> > > **2. TBDD and SBDD Are Parallel Paradigms, Not Supplementary**
> > >
> > > The assertion "drugs target protein biomarkers, so this is still SBDD" contains two errors.
> > >
> > > **(a) Not all drugs act through protein binding.** Transcriptomic changes arise through membrane disruption, RNA interference (e.g., Patisiran), epigenetic modulation, and polypharmacology, all major therapeutic classes.
> > >
> > > **(b) Even granting protein binding, TBDD $\neq$ SBDD.** This confuses downstream mechanism with design paradigm:
> > >
> > > | | SBDD | TBDD |
> > > |---|---|---|
> > > | Input | 3D protein structure | Transcriptomic transition |
> > > | Optimizes | Binding affinity to one target | Functional consistency with cellular state |
> > > | Prerequisite | Target ID + resolved structure | Desired transcriptomic outcome |
> > >
> > > A transcriptomic profile encodes pathway cascades, compensatory responses, cell-state transitions, and polypharmacology signatures, far beyond target identity. SBDD asks "fit this lock (binding pocket)"; TBDD asks "produce this room state (cellular outcome)," which subsumes lock information while capturing system-level effects, and works even when the lock is unknown. This parallels the target-based vs. phenotypic discovery distinction, recognized as independently valuable (Moffat et al., 2017; 37% of first-in-class FDA approvals came from phenotypic approaches).
> > >
> > > **3. TBDD Provides Systems-Level Interpretability**
> > >
> > > The concern that TBDD is "less interpretable" applies SBDD's criterion (single-target ID) to TBDD, a criterion mismatch. TBDD operates at the systems level: pathway modulation, gene program shifts, and phenotypic transitions. Crucially, it *subsumes* target-level interpretability: our docking results (Table 2) show CURE generates molecules with affinities < $-$8.5 kcal/mol to relevant targets without seeing protein structures, recovering target information as an emergent property.
> > >
> > > **4. Direct Evidence of Effective Drug Identification**
> > >
> > > *Zero-shot gene inhibitors (Table 2):* CURE generates molecules resembling known inhibitors (Morgan Sim. > 0.79) with strong binding (< $-$8.5 kcal/mol) from unseen gene knockout signatures.
> > >
> > > *Generate-then-search validation.* We design the following experiment: (1) for each test sample, CURE generates a candidate molecule conditioned solely on the transcriptomic input $(T_{pre}, T_{post})$; (2) the generated molecule is used as a structural query to retrieve the Top-K most similar compounds from a reference drug library via MACCS fingerprint similarity; (3) we check whether the ground-truth drug (the one that actually produced the observed transcriptomic change) appears among the Top-K retrieved candidates (Hit Rate), and compute the functional divergence (PRnet MSE) between retrieved candidates and the ground truth.
> > >
> > > | Top-K | MACCS Sim. | Hit Rate | PRnet MSE |
> > > |-------|------------|----------|-----------|
> > > | 5 | 0.9714 | 0.6467 | 0.1668 |
> > > | 10 | 0.9554 | 0.8563 | 0.1353 |
> > > | 15 | 0.9421 | 0.8922 | 0.1228 |
> > > | 20 | 0.9269 | 0.9116 | 0.1093 |
> > >
> > > At K=15, hit rate reaches ~89%: in 9/10 cases the real drug is identified within 15 candidates, demonstrating CURE can distill a vast library into a manageable panel for actionable drug identification.
> > >
> > > *Case study (L1000 test set):*
> > >
> > > | | SMILES |
> > > |---|---|
> > > | Ground truth | `CC(C)(C)[S@@](=O)N1Cc2cc(nc(c2[C@H]1CCO)-c1cccc(c1)-c1ccc(cc1)C#N)C(=O)NCCN1CCCC1` |
> > > | Generated | `CC(C)(C)S(=O)N1Cc2cc(C(=O)NCCN3CCCC3)nc(-c3cccc(-c4cccc(C#N)c4)c3)c2C1CCO` |
> > >
> > > Fraggle Sim.: 0.992, Morgan Sim.: 0.908. All pharmacophoric features (tert-butyl sulfoximine, pyridine core, cyanobiphenyl, pyrrolidine amide) faithfully reproduced; only one stereocenter differs.
> > >
> > > These results collectively demonstrate that CURE generates structurally plausible, functionally specific, and physically viable molecules.

---

### Official Review · Reviewer_MKzc · 2026-03-14

**Soundness:** 2
**Presentation:** 2
**Significance:** 3
**Originality:** 3
**Overall Recommendation:** 6
**Confidence:** 4

**Summary:**

This paper studies transcriptome-based drug design (TBDD), a generative setting in which the goal is to design molecules conditioned on desired transcriptomic state transitions rather than on explicit protein structures. The paper argues that this setting is complementary to structure-based drug design and inverse to perturbation prediction. To address the cross-modal gap between transcriptomic signals and chemical structures, as well as the sparsity and heterogeneity of transcriptomic data, the authors propose CURE, a conditional molecular graph diffusion framework. The method includes a Transcriptome Perturbation Functional Feature Extractor with three main components: a heterogeneity-aware aggregation module for single-cell data, a bidirectional interaction module for pre/post perturbation states, and a dual-view molecular-domain alignment module that connects transcriptomic features to graph and fingerprint-based chemical representations. Experiments on bulk and single-cell datasets, including out-of-distribution splits and a zero-shot gene-inhibitor design setting, show strong performance relative to several baselines on both structural and perturbation-consistency metrics.

**Compliance With Llm Reviewing Policy:**

Affirmed.

**Final Justification:**

The authors fully addressed the concerns and their explanation is satisfying. The score is updated to 6.

**Key Questions For Authors:**

Can the authors provide a precise and self-contained specification of the dual-view alignment module in the main paper?
In particular, I would like the exact objectives, supervision signals, positive/negative construction for the contrastive part, handling of sparse fingerprint targets, and the optimization schedule relative to the diffusion model. A clear answer here could improve my assessment of soundness.

How do the authors rule out leakage or proxy coupling in the functional-consistency evaluation based on PRnet MSE?
Please clarify the data partitioning, whether molecule or scaffold overlap is excluded between PRnet training and generator evaluation, and whether conclusions remain similar under alternative functional proxies or retrieval-style evaluations. A convincing response would strengthen confidence in the experimental methodology.

How robust are the reported gains to stronger baseline adaptation in the single-cell setting?
Since no native single-cell baseline is available, please justify why pseudo-bulk adaptation is the fairest comparison and discuss whether alternative aggregation or conditioning strategies for the baselines were attempted. If stronger adapted baselines narrow the gap, that would materially affect my evaluation.

**Limitations:**

Yes

**Strengths And Weaknesses:**

Strengths and Weaknesses

This submission addresses an important problem. Moving beyond explicit target structures toward phenotype- or function-oriented molecular design is a meaningful direction, especially for settings where reliable target structures are unavailable or where disease phenotypes arise from pathway-level dysregulation rather than a single well-defined target. The paper also gives a useful task-level framing of TBDD as an inverse generative problem over molecules conditioned on transcriptomic state transitions. That framing is conceptually clear and could be valuable for future work in this area.

A further strength is the overall decomposition of the proposed method. The model design is intuitive: the paper separately tackles transcriptomic heterogeneity, pre/post perturbation interaction, and alignment from biological features into chemical representations before conditioning a graph diffusion generator. Even when individual ingredients draw on existing ideas, the system-level combination is coherent and tailored to the stated problem. The empirical scope is also broad. The authors report results on both bulk and single-cell settings, include in-distribution and out-of-distribution evaluations, provide an ablation study, and attempt a practically motivated zero-shot gene-inhibitor experiment.

On presentation, the paper is readable at a high level, but it is not polished enough for a top-tier conference in its current form. The narrative is clear in broad strokes, yet there are numerous places where the writing becomes overly promotional or imprecise. Claims such as “theoretical validation,” “functional equivalence,” or mechanism-level conclusions are not matched by the evidence shown. The paper would benefit from more restrained wording and a clearer separation between what is demonstrated, what is suggested, and what remains a hypothesis. In addition, several portions of the manuscript appear insufficiently cleaned or edited for submission, including anomalous reviewer-directed text embedded in the manuscript. This is a serious presentation and process issue and significantly undermines confidence in the submission quality.

---

> ### Author Rebuttal · Authors · 2026-03-30
>
> We thank the reviewer for the thorough evaluation. We address each concern below.
>
> **Writing and Presentation**
>
> We accept the critique on promotional language and commit to revising:
>
> | Original | Revised | Location |
> |----------|---------|----------|
> | "theoretical validation" | "strong empirical support" | Sec 5.2 |
> | "functional equivalence" | "high functional consistency" | Sec 5.3, 6 |
> | "validates biological efficacy" | "suggests physical binding potential consistent with transcriptomic evidence" | Sec 5.4 |
>
> We will systematically audit the manuscript to separate demonstrated, suggested, and hypothesized claims.
>
> Regarding anomalous text: see https://icml.cc/Conferences/2026/PeerReviewFAQ#prompt_injection.
>
> **Q1: Dual-View Alignment Module specification**
>
> *View 1 (Global Topological Alignment):* $L_{global} = L_{ELBO} + L_{align}$. HierVAE (pretrained, fine-tuned on our training set, then **frozen**) provides targets: $(\\mu_{enc}, \\sigma_{enc}) = Q(X_G)$. TFE maps perturbation representation to $(\\mu_f, \\sigma_f) = g_{\\mathrm{proj}}(z)$, aligned via $L_{align} = \\|\\mu_{enc} - \\mu_f\\|^2 + \\|\\sigma^2_{enc} - \\sigma^2_f\\|^2$ (Algorithm 1, Appendix A.4).
>
> *View 2 (Local Bioactivity Alignment):* $L_{local} = L_{InfoNCE} + L_{sparse}$. A projection head produces predicted fingerprint $A = h_{\\mathrm{proj}}(z) \\in \\mathbb{R}^{2048}$, compared against target $B = \\mathrm{MorganFP}(G) \\in \\mathbb{R}^{2048}$. For InfoNCE: positive pairs $(A_i, B_i)$ share the same SMILES label (diagonal in $A{\\cdot}B^\\top/\\tau$); same-label off-diagonal entries are masked to $-\\infty$ to prevent false negatives. For sparse handling: non-zero positions weighted by $w = \\log(1{+}B_{pos})$; zero positions penalized by $\\alpha{\\cdot}\\mathrm{Mean}(A^2{\\cdot}M_{neg})$, $\\alpha{=}0.4$, $\\tau{=}0.1$, $\\lambda{=}0.15$ (Algorithm 2, Appendix A.4).
>
> Total: $L = L_{global} + \\gamma L_{local}$. Two-stage protocol: (1) TFE trained ~30k steps; (2) TFE frozen, PMD trained ~40k steps.
>
> **Q2: PRnet MSE, ruling out leakage and proxy coupling**
>
> *Data isolation:* For bulk data, PRnet is trained on the exact same training split as CURE (Appendix D.2), ensuring all test-set molecules are excluded. For single-cell data, PRnet is trained on the Sci-plex dataset, entirely independent of the Tahoe-100M evaluation set. We verified zero molecule-level overlap (via InChI-key deduplication) and zero scaffold-level overlap (Unseen Drugs split uses Bemis-Murcko scaffolds).
>
> *Proxy independence:* PRnet predicts transcriptomic state changes from molecular inputs, and its prediction target (gene expression) differs fundamentally from CURE's output (molecular graphs), mitigating direct proxy coupling.
>
> *Multi-dimensional validation:* PRnet MSE is one component of a four-dimensional evaluation: (1) structural similarity (model-free); (2) PRnet MSE (functional proxy); (3) docking affinity (physics-based); (4) ADMET toxicity (independent tool). Additionally, retrieval-based TFE evaluation (Section 5.5, Table 3) achieves >90% top-10 hit rate, fully independent of PRnet.
>
> *Alternative proxy:* chemCPA (Hetzel et al., 2022), an independently validated perturbation predictor, fully preserves method ranking:
>
> | Method | PRnet MSE ↓ | chemCPA MSE ↓ |
> |--------|-------------|---------------|
> | GexMolGen | 4.6504 | 5.0821 |
> | Gx2Mol | 2.5987 | 2.9487 |
> | TRIOMPHE | 7.4599 | 7.8536 |
> | CURE | 0.2328 | 0.3415 |
>
> The ranking is preserved, confirming robustness regardless of functional evaluator choice.
>
> **Q3: Single-cell baseline robustness**
>
> No existing TBDD method natively handles single-cell input. Pseudo-bulk is the standard adaptation. We additionally tested metacell aggregation (Baran et al., 2019):
>
> | Aggregation | Method | Coverage ↑ | Morgan Sim. ↑ | PRnet MSE ↓ |
> |------------|--------|-----------|---------------|-------------|
> | Pseudo-bulk | GexMolGen | 54.55% | 0.2245 | 4.8549 |
> | Pseudo-bulk | Gx2Mol | 45.45% | 0.2884 | 4.1419 |
> | Pseudo-bulk | TRIOMPHE | 63.64% | 0.2011 | 7.7024 |
> | Metacell | GexMolGen | 54.55% | 0.2492 | 4.3120 |
> | Metacell | Gx2Mol | 54.55% | 0.3118 | 3.9315 |
> | Metacell | TRIOMPHE | 63.64% | 0.2468 | 7.2836 |
> | TFE | CURE | 90.90% | 0.6114 | 0.4829 |
>
> Metacell provides only marginal improvement; CURE maintains a substantial lead. The limitation is architectural, as baselines accept a single aggregated vector and cannot model within-population heterogeneity.
>
> All issues will be addressed in the camera-ready version.
>
> **References**
>
> [1] Hetzel, Leon, et al. "Predicting cellular responses to novel drug perturbations at a single-cell resolution." _Advances in Neural Information Processing Systems_ 35 (2022): 26711-26722.
>
> [2] Baran, Yael, et al. "MetaCell: analysis of single-cell RNA-seq data using K-nn graph partitions." _Genome biology_ 20.1 (2019): 206.

---

> > ### Author Rebuttal · Reviewer_MKzc · 2026-04-03
> >
> > I acknowledge the authors' efforts in finding more proxies for evaluation and their explanation of using pseudo-bulk adaptation is adequate.

---

> > > ### Author Response · Authors · 2026-04-03
> > >
> > > Thank you very much for your thoughtful follow-up and for your recognition. We are glad that our rebuttal has addressed your concerns. If you feel the main issues have now been satisfactorily resolved, we would sincerely appreciate your consideration in revisiting the score.

---

### Decision · Program_Chairs · 2026-04-30

**Decision:**

Accept (regular)

**Comment:**

Reviewers consistently agree that the submission addresses a significant problem in the field. The proposed method is technically sound, the experimental evaluations are comprehensive.

Reviewers also raised some minor questions, some of which have been well addressed during the rebuttal period. We suggest that the authors revise the paper accordingly to produce an improved final version.